# CARRYING OVER ALGORITHM IN TRANSFORMERS

## ABSTRACT

Addition is perhaps one of the simplest arithmetic tasks one can think of and is usually performed using the carrying over algorithm. This algorithm consists of two tasks: adding digits in the same position and carrying over a one whenever necessary. We study how transformer models implement this algorithm and how the two aforementioned tasks are allocated to different parts of the network. We first focus on two-layer encoder-only models and show that the carrying over algorithm is implemented in a modular fashion. The first layer is mostly responsible for adding digits in the same position. The second layer first decides, in the attention, which positions need a carried one or not, and then performs the carrying of the one in the final MLP. We provide a simple way of precisely identifying which neurons are responsible for that task. This implementation of the carrying over algorithm occurs across a range of hyperparameters for two as well as three-layer models. For small decoder-only models, we observe the same implementation and provide suggestive evidence for its existence in three 7B large language models.

## 1 INTRODUCTION

While large language models (LLMs) continue to shown fascinating capabilities across many different modalities and tasks Anil et al. (2023); Touvron et al. (2023a;b); OpenAI (2023), their mathematical reasoning abilities seem to be lagging behind. Various direction on how to improve this have been explored, for instance, using carefully selected data Lewkowycz et al. (2022); Azerbayev et al. (2023) or different inference strategies Imani et al. (2023). Nevertheless, at its core, mathematics is about various logical implications and algorithms that have to be employed in order to perform well on a given task. It therefore seems natural to ask how LLMs implement such algorithms. By itself this is a difficult endeavour because of the complexity of these models, but one fruitful approach is to first study smaller, interpretable toy models and apply the lessons learned there to the larger, more complex models Elhage et al. (2022a); Olsson et al. (2022); Elhage et al. (2022b).

In this work we utilize this approach for understanding integer addition in transformer based models. Working with the digit representation of integers, a natural algorithm that could be implemented by the model is the carrying over algorithm. We study how this algorithm is implemented in transformer models ranging from one to three layers, focussing first on encoder-only architectures, but also elaborate on decoder-only models. We then try to apply our findings to understand length generalisation for integer addition and integer addition in the fine-tuned LLMs Alpaca 7B Taori et al. (2023), Llemma 7B Azerbayev et al. (2023) and Zephyr 7B Tunstall et al. (2023).

The combination of the dataset and the simplicity of the carrying over algorithm, provides a rich playground to test and interpret various aspects. There are two reasons for this. First, the algorithm combines two tasks: addition of the digits within each integer and carrying of the one. This gives ample insight into the models' performance. Second, the dataset has natural subsets, depending on where a carried one is to be added or not, which can be used to assess the model's performance and inner workings at a more fine-grained level.

Intuitively speaking, the reasons why one can expect the transformer architecture to implement at least part of the carrying over algorithm is due to the self-attention. In the digit representation, once the digits are converted to vectors in the latent space, the operations one would like to perform are addition of vectors at different positions. The self-attention mechanism makes that possible once the correct pattern has been learned.

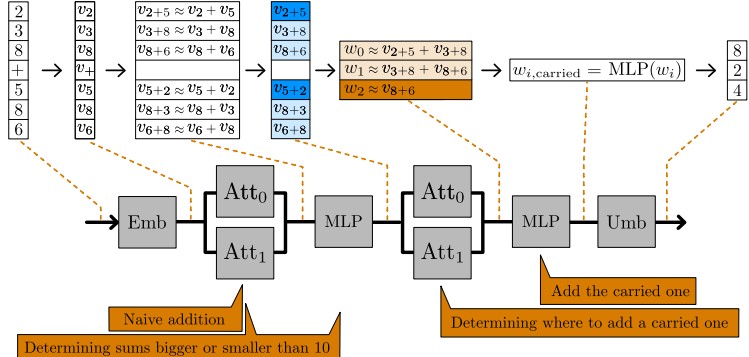

Figure 1: Summary of two-layer models' implementation of the carrying over algorithm. Note that when we write the addition of two vectors, we mean a linear combination, but for clarity we did not write the coefficients. The light blue indicates $\geq 10$ and darker $< 10$. Similarly, the light orange indicates that a carried one needs to be added, whereas for the dark orange it is not necessary.

## 1.1 OUR CONTRIBUTIONS

To stipulate our contributions more clearly, we think of the carrying over algorithm as consisting of four steps: 1) Add digits at each position of each integer. 2) Determine whether resulting sums are bigger or smaller than 10. 3) Determine where a carried one needs to be added. 4) Add carried one.

1. For two-layer encoder-only models, each step of the carrying over algorithm is implemented in a parallel fashion by a particular part of the architecture, see Fig. 1 for a summary of the 3 digit addition case. For 4 digit addition we reach the same conclusion, see App. F. We find the same implementation in three layers as well as decoder-only models, see App. C.3 and D.

2. Some of the lessons learned for smaller models carry over to fine-tuned LLMs, Alpaca 7B Taori et al. (2023), Llemma 7B Azerbayev et al. (2023) and Zephyr 7B Tunstall et al. (2023). We provide suggestive evidence for this.

3. The implementation of the carrying over algorithm for short length (3 digits) generalizes to larger ones after priming Jelassi et al. (2023) the training with a tiny set of larger addition sums. With finetuning we also achieve similar generalisation.

4. One-layer models experience a novel phase transition in the models learning dynamics, where the QK circuit suddenly changes and the attention blocks become adders. For larger models this transition happens very early on in training.

## 1.2 RELATED WORKS

Considerable effort has been devoted to understanding if and when transformers trained on algorithmic tasks can generalize to out-of-distribution examples, e.g. length generalisation for integer addition. For instance, Csordás et al. (2021) considers modifications to the standard transformer architecture, whereas Jelassi et al. (2023); Kazemnejad et al. (2023) focus on different positional embeddings. The input format is also crucial for good performance as shown by Nogueira et al. (2021); Lee et al. (2023). From a more computational point of view, using RASP Weiss et al. (2021), the abilities and limitations of transformers has been studied in Zhou et al. (2023). On the interpretability side, there has also some recent interest into reverse-engineering small transformers trained on addition tasks, see for instance Nanda et al. (2023).

## 2 SET-UP

**Dataset** For concreteness we focus on three digit addition, but see App. F for four digit addition. In Sec. 6 we also study generalisation to six digit, using priming Jelassi et al. (2023). Our dataset is constructed using positive integers $a, b$ such that $a + b < 1000$ and we tokenize at the digit level. This means we split $a, b$ as $a \rightarrow a_0 a_1 a_2, b \rightarrow b_0 b_1 b_2$ with each digit a separate token. We also add

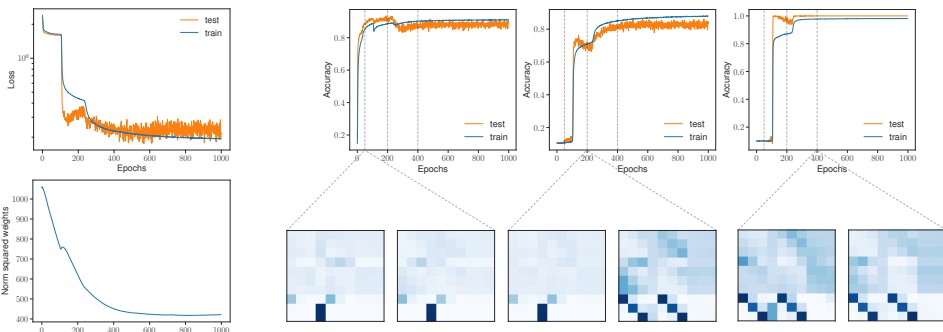

Figure 2: **Left:** Loss and norm of the weights as a function of epochs for both the training and test data. Train/test split is $s = 0.3$ and $\lambda = 0.2$. **Right:** Attention pattern for each head at epoch 50, 200 and 400. There is a distinct pattern after each transition (we checked the transition is indeed sudden), which can happen separately in each head and has the structure so as to add embedding vectors and transfer them to the output positions. The attention patterns are averaged over the test dataset.

a token + in between the two integers and attach three = at the end, which we use to read out the model's output. This results in a vocabulary of size 12 and sequence length 10.

As mentioned above, the beauty of this dataset is its natural division in subsets depending on: non-carrying over sums and the carrying of the one. This division will be an integral part of our discussion and is important to keep in mind. For three-digit addition there are 5 such tasks (subsets):

1. Non-carry (`NC`): $a_i + b_i < 10 \; \forall i$

2. Sum is larger than 10 only at position 1 (`C@1`): $a_1 + b_1 \geq 10, a_i + b_i < 10$ for $i \neq 1$

3. Sum is larger than 10 only at position 2 (`C@2`): $a_2 + b_2 \geq 10, a_1 + b_1 < 9$

4. All but the first sums $\geq 10$ (`C all`): $a_i + b_i \geq 10$ for $i \neq 0$

5. Outer most sum $\geq 10$ but second to last equal to 9 (Consecutive carrying over, `C all con.`): $a_2 + b_2 \geq 10, a_1 + b_1 = 9$

We will also distinguish examples according to the digit at the last three positions, which reflects the carrying of the one. To avoid confusion, we will count the positions of the digits from the left.

**Models & Training**  In the main text we consider either one or two layer transformer models Vaswani et al. (2017); Devlin et al. (2018) with LayerNorm and dropout ($p = 0.1$). The model dimensions are $d_{\text{model}} = 128$ and $d_{\text{ff}} = 128$ (although we also consider larger $d_{\text{ff}}$). Furthermore, we consider models with two heads, and used the RoFormer for positional embedding Su et al. (2021). In App. C we consider other hyperparameters and discuss a three layer model. These models and dataset size (500500 examples) puts us in the overparametrized regime. With the four digit case considered in App. F, we will be in the underparametrized regime.

The models are trained on the (shuffled) dataset with a train/test split $s = 0.3$ and weight decay $\lambda = 0.2$. We use a constant learning rate $\eta = 1.4 \times 10^{-4}$, a batch size of 1024 and train for 1000 epochs with AdamW. We consider sets of six trained models with different random initializations. See App. A for more details. While we mostly focus on encoder-only models, we also discuss generative models in App. D and reach similar conclusions as the ones presented below.

**Methodology**  To reverse engineer the aforementioned models, we employ two strategies:

1. We judge the importance of parts of the model based on the per token accuracy (instead of the loss) before and after (zero) ablation and combine this with the 5 tasks discussed above. This division allows us to clearly distinguish which parts are necessary for what tasks and will give a fairly precise understanding of the models' performance.

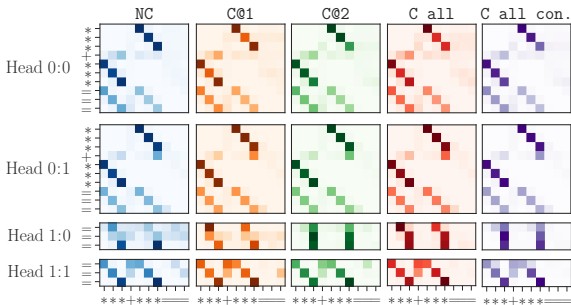

Figure 3: Attention pattern for each head and layer for a particular run (of the six). Each column represents one of the five tasks (see Sec. 2). For the last layer we only plotted the three output positions (=). Again we see the staircase patterns for an interaction between the digits (∗) of each integer. Furthermore, in head 1:0 we see how information from the previous sum gets transferred to the current sum so as to determine whether a carried one is needed or not. It is slightly different for each column in the way one expects. For instance, in the third column, the second position of the outcome gets attention from the sum of digits of the last position of each integer.

2. We study how residual stream gets transformed by the model and perform a PCA on the output of the attention and MLP to gain mechanical insight into the model.

We found the actual embedding and unembedding to not be very interesting, so we will refrain from discussing them in the main text. A short discussion can be found in App. C.

## 3 ONE LAYER

**Phase transitions in the Attention**  To set the stage, we first consider one-layer models. Fig. 2 shows the loss, accuracy and the weight norm as a function of epochs for one of the six runs. The one-layer models do not reach perfect accuracy, but experience a phase transition where suddenly the models loss and accuracy improve and weight norm bounces Lewkowycz (2021). This non-grokking transition is driven by a phase transition in the QK-circuit Olsson et al. (2022) in the attention block. In particular the attention pattern suddenly changes for the second and third position by forming a *staircase* pattern, as can be seen in the right panel of Fig. 2.

These *staircase* patterns after the transitions are actually very intuitive. Not only do they provide the necessary attention between the digits one wants to add (and hence the drop in loss), but also perform a literal (naive) addition of the embedding vectors for each position in parallel. It is naive, as it will distinguish between e.g. $2 + 1$ and $3 + 0$, which the rest of the model needs to correct for.

**MLP**  The one-layer models are not perfect at 3-digit addition and as a result are harder to fully reverse engineer, especially the MLP. In App. B we will study these models in some more detail. In the two-layer models to be discussed below, there is much more structure beyond just the attention patterns and in particular, the final MLP has a dedicated functionality that will help us understand the mechanics of these models much better.

## 4 TWO LAYERS

Let us now consider two layer models. In this section we will show that the carrying over algorithm[1] is implemented in a modular fashion according to the steps shown in Fig. 1. An interesting analogy with an electrical circuit is discussed in App. I. We will consider one single run (the same every time) or the average over all six. All models reach perfect accuracy and determine the *staircase* attention pattern quickly. They do not experience interesting behaviour in their learning dynamics, so we will move on to interpret the trained model, but see Fig. 11 for the learning dynamics.

---

[1] We will use 'algorithm' throughout the text, although strictly speaking, as we will see in Sec. 6, it will not generalize to larger length without finetuning or priming Jelassi et al. (2023).

Table 1: Accuracy after ablating the *decision* head and final MLP for the 5 tasks, averaged over six runs. The 'corrected' accuracy for non-carry sums are obtained by manually subtracting one from each position of the output of the model and comparing that with the target. This reveals how many of the non-carry sums got an incorrect carried one. For carrying over sums we added a one so as to see for what example the model forgot to add a carried one. For ablating the decision heads: whenever it is unsure, the variation between runs is large ($\sim 0.38$).

| Task | *Decision* head | | | Final MLP | | |
|---|---|---|---|---|---|---|
| | pos. 7 | pos. 8 | pos. 9 | pos. 7 | pos. 8 | pos. 9 |
| NC | 0.52 | 0.31 | 1.0 | **0.90** | **0.95** | **0.96** |
| Corrected NC | 0.48 | 0.69 | 0.0 | 0.10 | 0.05 | 0.04 |
| C@1 | 1.0 | 0.49 | 1.0 | 0.14 | **0.99** | **0.96** |
| Corrected C@1 | 0.0 | 0.51 | 0.0 | **0.86** | 0.01 | 0.04 |
| C@2 | 0.58 | 1.0 | 1.0 | **0.89** | 0.10 | **0.99** |
| Corrected C@2 | 0.42 | 0.0 | 0.0 | 0.11 | **0.90** | 0.01 |
| C all | 1.0 | 1.0 | 1.0 | 0.14 | 0.01 | **0.99** |
| Corrected C all | 0.0 | 0.0 | 0.0 | **0.86** | **0.99** | 0.01 |
| C all con. | 1.0 | 1.0 | 1.0 | 0.12 | 0.0 | **0.99** |
| Corrected C all con. | 0.0 | 0.0 | 0.0 | **0.88** | **1.0** | 0.01 |

**Attention**   The attention blocks are again adders and transfer information from the relevant digits to each other. The attention patterns are shown in Fig. 3, where each column represents one of the five tasks discussed in Sec. 2, and the rows correspond to the layer and head. Let us focus on the first layer. We see that the first seven rows are for attention between the digits of each integer. This *staircase* pattern is important for transferring information from one digit to another and although it makes the attention itself a copier, the inclusion of the residual stream (skip connections) makes it into an adder. This suggests the skip connections are crucial for it to add and indeed the average accuracy (over six runs) drops to $0.13 \pm 0.1$ if we remove the information in the skip connection.

For the second layer, we see that head 1:0 transfers information from the previous sum to the current sum. For instance, for C@1 we see that the first row receives attention from the first and fifth position. These contain the information from the sum at the second position of each integer, obtained from the first layer. The information of the sum relevant for the first position is then transferred by head 1:1 and the skip connections. This is the models' implementation for determining whether a one needs to be carried or not, but, from what we have seen in other runs, this is typically not the only path.

To understand these pathways, we study the effect of ablating the heads in the second layer. We found there to always be one head that was more important than the other. We call this head a *decision* head because it turns out to help decide whether or not to add a carried one. The accuracies for the five tasks after ablating the decision head are shown in Tab. 1. There are two interesting observations: 1) The ablated model is very confident where to add a one when it indeed should. 2) when it is not supposed to add a one, it *either* adds the one *or* it does not, i.e. the accuracy is exactly divided between correct or off by one. This suggests the decision heads are necessary to make a proper decision on where a carried one is needed. In the next subsection we will discuss another piece of evidence for this behaviour.

**MLP**   Despite the attention making the decision, the final MLP actually adds the carried one. To show this, we performed two experiments: 1) ablate the MLP, 2) study its action on sums with a fixed outcome. The results for the former experiment are shown in Tab. 1. We see the ablated model can perform all calculations almost perfectly, except at positions where carrying over is necessary. We added or subtracted a one for the non-carry or carry sums, respectively, to see whether it simply forgot to add a carried one. The results indeed confirm this and suggest that the final MLP is responsible for adding the carried one.

Furthermore, we saw that ablating the decision heads, the model becomes unsure about non-carry sums at positions 7 and 8 and sometimes adds a one. If the MLP indeed carries the one whenever necessary, it should also mean that when we ablate it, the model should become very good again at non-carry sums, but bad at everything else. This is indeed what we found.

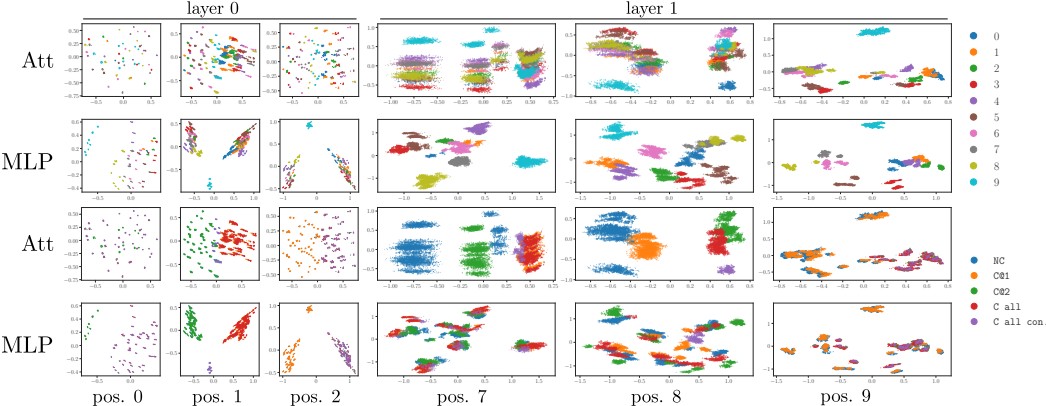

Figure 4: PCA for the outputs of the attention and MLP blocks in each layer for the two leading principal axes. First three columns are the first layer and positions 0, 1 and 2 (other positions are not shown as they are similar), the last three columns are the second layer at the positions of the outcome. For rows 0 and 1: layer 0 plots are labelled according to the sum (ignoring any carried one) at that position, layer 1 plots according to the answer at that position. For rows 2 and 3 we labelled according to the tasks discussed in Sec. 2. We see the first layer determines whether sum $< 10$ or $\geq 10$ and groups those examples (separating also the 9 as a special case). The second layer instead groups examples according to whether they need a carried one or not. Notice that position 9 never needs a carried one and so the examples are grouped in the same way as in the first layer.

For the second experiment we study the action of the final MLP on embedding vectors for sums with the same outcome. The model should rotate those embedding vectors towards each other, i.e. squashing them together. Averaged over six runs, we find this to be the case. The squashing[2] is $0.42, 0.13$ and $0.28$, with standard deviation $0.20, 0.02$ and $0.11$ for position $7, 8$ and $9$.

## 4.1 A JOURNEY OF HIDDEN REPRESENTATIONS

In the previous subsections we found evidence that the model implements the carrying over algorithm in a rather modular way. To provide more evidence and get a more detailed understanding of what those modular parts are doing, it is useful to consider the model's residual stream. We consider a set of $20k$ random examples and perform a PCA after the attention and MLP of each layer. For the two leading principal axes this resulted in Fig. 4. Lets unpack this figure.

**Layer 0: Determining whether sum $< 10$ or $\geq 10$.** The first two rows display the attention and MLP (resp.) output labelled according to the outcome ignoring any carrying. The last two rows are labelled according to the 5 tasks. We saw that the attention is acting like an adder, which manifests itself in the $2d$ projection by the leading axis corresponding to whether the sums at the input integers' digits are $\geq 10$ or not. E.g., at position 2, NC and C@1 (which have sums $< 10$) have clearly separated themselves from the other examples that have a sum $\geq 10$. After the MLP these groups are more localized and the subleading direction (for positions 1 and 2) now distinguishes when a sum (at positions $i$ and $i + 4$) resulted in a nine or not. It makes sense that a nine is treated differently as it can only arise from non-carrying over sums.

**Layer 1: Determining whether a carried one needs to be added and adding it.** The labelling for rows is the same as for layer 0, but we did include any carried one for the first two rows. We already know that the attention in the second layer transfers the information in the positions of the the two digits to the outcome positions. This results in a clear separation between the 5 tasks along the leading principal axis and an (rough) ordering of the digits along the first subleading axis at positions 7 and 8. At position 7 we see non-carry sums (NC and C@2) are separated from the ones that require a carried one at that position (C@1, C all and C all con.). The same is true for

---

[2]Defined as the ratio between the difference in maximum and minimum of the overlap before and after the MLP. A ratio $< 1$ means the vectors are squashed together.

position 8. For position 9 there is no carrying and so the structure is similar to the output of the previous layer. These observations also supports our earlier observation that in the second layer one of the heads is responsible for *deciding* whether a one needs to be added later. [3] This job, as we saw, is performed by the final MLP. The outputs of this block are not grouped according to task, but rather through their outcome; each integer roughly has its own localized region in the $2d$ projection.

In some cases, one of the two leading principal directions did not correspond to whether a carried one is needed or not. For those cases we could find a more subleading one that was. Further structure in the residual stream (such as a pentagonic arrangement of the inputs) is discussed in App. E.

## 5 A CLOSER LOOK AT THE FINAL MLP

We argued that the final MLP is responsible for adding the carried one. How does this specialization develop over the course of training and are all neurons necessary for that?

**Dissection**  In general we found the answer to the last question to be *no*. Finding the subset of neurons that *are* important is not easy, however. The task could not be aligned with the activations Elhage et al. (2022b;a) or because part of the calculation is done elsewhere in the network. This calls for a more systematic approach. For instance, by studying the activations or do an SVD on the pre-activation weights to determine what neurons are most relevant, see e.g. Voss et al. (2021).

In the former case we found that ablating those neurons whose activations statisfy $z_i > z_{\mathrm{NC}}$ where $i$ is any of the five tasks that requires carrying over, has profound effects on *just* the carrying over capabilities and *not* the non-carry over sums. For the two-layer model discussed thus far, this procedure results in a set of $\sim 86$ ablated neurons that cause a *total* inability to carry the one, i.e. the accuracy is 1 on sums that do not require carrying over and 0 otherwise, but in which case the corrected accuracy is 1. In App. C.4 we employ this protocol on models with other hyperparameters, including those where ablating the entire MLP caused the accuracy to drop on all tasks.

Using the SVD approach we can associate the leading axis with the feature *carrying over*. The value along that axis determines how important the neuron is for carrying over. For the model discussed so far, see Fig. 14. See also Fig. 13.

Mechanistically, the ablated neurons rotate the hidden reps of each task towards the (correct) output. This can be made more explicit by giving the ablated model a set of examples with fixed outcome. The matrix of cosine similarities between the hidden reps then follows a checkerboard pattern (following the carrying over nature of the sum) instead of roughly equal to 1 everywhere.

**Evolutions**  Besides looking at the trained MLP, we also considered its training development and how that relates to features in the loss. We do this by ablating the final MLP of the trained model after every epoch and record the accuracy (corrected and non-corrected) for each of the five tasks. We then determine the Pearson correlation coefficient (PCC) with the pattern of accuracies in the situation for which the ability of carrying the one is removed entirely. The resulting dynamics is shown in Fig. 5.

We see the final MLP's function already starts developing quite early on in the training. If we look at the test loss, there is a kink which coincides with the moment the PCC starts increasing or the non-corrected accuracy in the second panel of Fig. 5 starts decaying. After this kink the test loss has a long period of exponential decay with approximately constant decay rate. Here we plotted only 400 of the 1000 epochs, but the PCC reaches 0.98 at the end.

The analysis performed here also highlights that the different tasks can be used to not only measure performance in a more fine-grained way at the end of training but also during. It can be used as a progress measure, but due to the exponential decay of the test loss it is not as hidden as was, for instance, discussed in Nanda et al. (2023); Barak et al. (2022).

---

[3]We confirmed this by ablating the *decision* head causing the separation between tasks to be absent. Instead examples from different tasks are now mixed together, creating the confusion noted in Tab. 1. For instance, examples in C@1 at position 8 will now overlap with C all, only C all needs a carried one there. Ablating the non-decision head does change the detailed 2d projection, but leaves the separation of tasks intact.

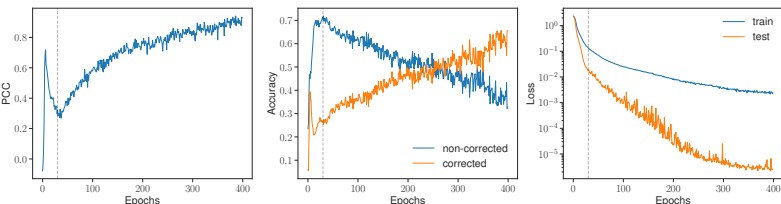

Figure 5: MLP evolution. **Left:** Pearsons correlation coefficients of accuracies (corrected and non-corrected) of ablated model with the accuracies expect when no carrying of the one can be performed. **Middle:** Accuracy for carrying of the one at position 7 (i.e. set `C@1`). Note the corrected accuracy is obtained by adding a one at position 7 to see if it 'forgot' to add a one. **Right:** Test/train loss. The dashed vertical lines indicates the kink discussed in the main text.

## 6   GENERALISATIONS

**Length generalisation**   A fundamental limitation of our discussion so far is that we just looked at 3 digit addition and not studied the behaviour on larger integers *not* seen in training. To study this, we need to add padding to our inputs to allow for additional positions and train new models. For instance, let us consider training on 3 digit and testing on 6 digit addition, keeping other hyper-parameters fixed. As expected, these models do not generalise, reaching only 10-15% accuracy on 6 digit sums. We will discuss a specific model in detail in App. H. The upshot of that discussion is actually that all the parts of the carrying over algorithm are in place around epoch 500 (and at which point the accuracy is relatively high), but after this time the model starts to forget this and focusses purely on doing three digit addition well. This instability seems to suggest one should *prime* the model with very few examples Jelassi et al. (2023). Indeed, following the prescription in Jelassi et al. (2023) with 100 six digit sums added to the 3 digit training set, we can train models with close to perfect addition capabilities on six digits. Performing the same deep dive as in Sec. 4, we find that the primed model indeed implements the carrying over algorithm, see App. H.1. By comparing to the unprimed model, we suspect that the reason why generalisation using a tiny set of priming examples works, is because it utilizes the carrying over components for just three digit addition developed early on in training.

One way to study that further is to see whether finetuning the unprimed model (at, say, epoch 500) would learn six digit addition relatively fast without changing the model's weights too much. We find evidence for this, see App. H.2. There we took only 500 six digit sums as our finetuning dataset and trained for 50 epochs. This caused the model to reach a test accuracy of 0.94 and 0.97 on six and three digit addition sums, respectively, but with tiny changes to the model weights.

**Large language models**   Returning to our initial (rather ambitious) motivation, let us now try to apply our lessons to LLMs. As already noted, LLMs are not great at integer addition, which might complicate finding clues for an implementation of the carrying over algorithm. Furthermore, due to the large number of ways the model could implement the algorithm and transfer information from token to token, a fully interpretable implementation is unlikely. A partial implementation seems more reasonable. Let us start by studying the attention patterns and the occurrence of the *staircases*. For this we prompted (zero-shot) Alpaca 7B Taori et al. (2023), Llemma 7B Azerbayev et al. (2023) and Zephyr 7B Tunstall et al. (2023); Jiang et al. (2023) to compute a set of $1k$ four digit addition sums[4]. For simplicity, the sums are such that the integers are in between $10^3$ and $10^4$. The results are shown in Tab. 2. Alpaca's accuracy is low, but Llemma's and Zephyr's accuracies are rather high, which, for Llemma, is not surprising as it is trained to do well on mathematics. We also found that 64% and 74% of Llemma's and Zephyr's mistakes are carrying over mistakes, respectively.

For these LLMs we found two heads to be of most importance, see Tab. 2 and App. A.3 for how we determined this. For Alpaca, these were heads 6:23 and 13:15. The former transfers information from the first to the second integer, whereas the latter provides the attention of the second integer to the output; similar to what we saw previously, but see also App. D. For Llemma 7B both *addition*

---

[4]We used the default Alpaca prompt, but prompted Llemma and Zephyr without any additional system message. See App. A.3 and A.4 for more details.

Table 2: Accuracy, number of heads with a (partial) staircase pattern and addition heads for Alpaca 7B, Llemma 7B and Zephyr 7B.

| model | acc. | # of heads with staircases | *addition* heads | ablated acc.[5] |
|---|---|---|---|---|
| Alpaca 7B | 0.33 | 164 | (6:23, 13:15) | 0.17 |
| Llemma 7B | 0.87 | 190 | (13:23, 15:15) | 0.30 |
| Zephyr 7B | 0.85 | 239 | (17:25, 17:26), (20:30, 20:31) | 0.34, 0.35 |

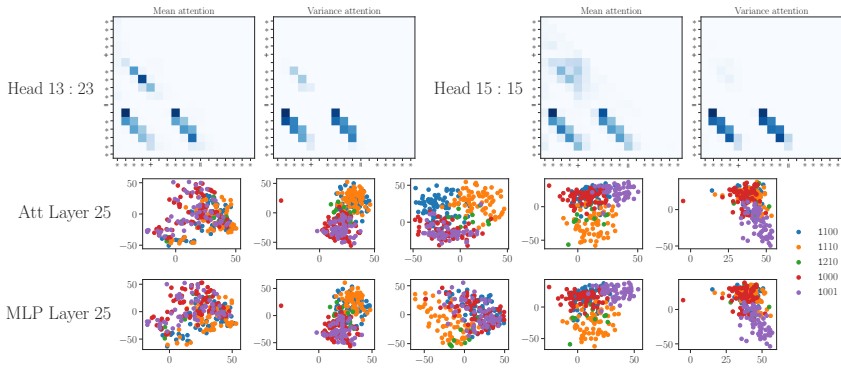

Figure 6: **Top:** Attention pattern of *addition* heads of Llemma 7B. We see the same staircase patterns as before. Mean and variance is over $1k$ examples. The variance is mostly located at the staircases. **Bottom:** Residual stream at layer 25. At the third position we see the MLP rearranging the hidde states according to whether a carried one is need or not instead of the value of the summed digits. Due to the large number of cases, we labelled them using a trinary system, where a 0 means $< 9$ at this position, 1 means $\geq 10$ and 2 means equal to 9 (if previous sum was $\geq 10$).

heads perform addition and information transfer to the output positions simultaneously. See Fig. 6 for the average attention patterns and their variance. Zephyr is similar to Llemma, but has more attention heads with staircase patterns. In Tab. 2 we give the two most important pairs of heads.

To get a more detailed understanding of how these LLMs do integer addition, we study the residual stream of the first five output positions of Llemma. It turns out this model establishes the same division between $\geq 10$ or $< 10$ and whether a carried one is needed or not, just as we saw before in Sec. 4. It will do so throughout the network, starting at around layer 15 and using different layers to manipulate the data from a division between $\geq 10$ or not to division between whether a carried one is needed or not, similar to the behaviour in Fig. 4. This is particularly visible for the third position in layer 25 as shown in Fig. 6. Zephyr has similar features of the small models' implementation of the carrying over algorithm. In App. G we consider Alpaca in some more detail.

These two observations of the attention patterns and residual stream having some similarities with the smaller models, potentially suggest that the modular implementation of the carrying over algorithm we found previously is perhaps also present in LLMs. With an eye towards improving LLMs' arithmetic abilities, it would be worthwhile to explore this further.

## 7 CONCLUSION

We studied small encoder-only transformer models trained on 3-digit addition. One-layer models showed a novel *non*-grokking phase transition in their QK-circuit, but did not learn addition perfectly. Two-layer models, on the other hand, implement the carrying over algorithm in a modular fashion with each step in the algorithm having a dedicated part of the network. With priming or finetuning these implementations generalize to six digit addition. We also looked at three LLMs as an application of our learned intuition and found typically 1) two relevant attention heads with the staircase patterns and 2) a residual stream having similarities with the ones found in smaller models.

---

[5]Importantly, this did not spoil the models' ability to generate responses in the same way as in the unablated model.

## REPRODUCIBILITY STATEMENT

In an effort to ensure reproducibility, we have explained in Sec. 2 what type of dataset was considered, how it was tokenized and what type of models and hyperparameters we used. More details can be found in App. A, which also includes details on the decoder-only models and LLMs we discussed in Sec. 6. Additionally, at `https://github.com/CarryingTransformers/CarryingTransformers.git` we provide the code for training the models discussed in the main text and appendices and jupyter notebooks to reproduce Fig. 2-4, 7-17, 20-25, Tab. 1, 3-5 and the claims in the main text. For Fig. 5 we saved the model after each epoch, so this too much to share, but it can be reproduced using the training code provided. For the figures relevant for App. F the dataset is too large to share, but again one can train a model with the code provided. In the repository we also provided the attention patterns and the PCA of the two leading axes of the residual stream of the three LLMs to support our claims.

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

APPENDICES

In the main text we referred to a couple of appendices, which can be found below.

## A  MODEL AND TRAINING DETAILS

### A.1  ENCODER-ONLY MODELS

Here we outline in more detail the model and training details. After the tokenization discussed in the main text, we use a one-hot encoding to embed our tokens in a 128 dimensional ($d_{\text{model}} = 128$) embedding space. After the embedding we have a couple of layers consisting of: LayerNorm, softmax self-Attention with RoFormer, LayerNorm and finally an MLP (of size $d_{\text{ff}}$). Around the first two and last two constituents of each layer we also have a skip connection. After the final layer we go through another LayerNorm and then we generate an output distribution with an unembedding and softmax layer. The prediction for the outcome is generated using greedy search.

Furthermore we note that except for the embedding matrix, no biases have been turned off[6] and that the MLP block has one hidden layer with a ReLU activation.

The models were trained using AdamW Kingma & Ba (2014) with parameters $(\beta_1, \beta_2) = (0.9, 0.98)$, $\varepsilon = 10^{-8}$ and learning rate $\eta = 1.4 \times 10^{-4}$ and weight decay $\lambda$. We used a mini-batch size of $1024$. The parameters (except for biases) are initialized using Glorot initialization.

### A.2  DECODER-ONLY MODELS

We used the same architecture as above, but treated the problem in a next-word prediction fashion. This changes how one computes the loss, which can be seen in the source code provided in the Github repository. As an end-of-sequence token we used an =-sign.

For training we also kept the same hyperparameters.

### A.3  ALPACA 7B DETAILS

We used the Alpaca repository to instruction fine-tune LLaMA-1 7B to obtain Alpaca 7B. For the training we used 4x40GB A100s and kept all their hyperparameters, except we had to use a per-device batch size of 2 instead of their 4 and employed deepspeed.

We constructed a set of 1000 examples of the form `a + b =` with $a$ and $b$ two random four-digit integers (this means $10^3 \leq a, b < 10^4$ so that it is easier to study the attention patterns). The prompts for the model are then generated using the Alpaca template:

```
"Below is an instruction that describes a task."
"Write a response that appropriately completes the request.\n\n"
"### Instruction: \n{EXAMPLE}\n\n### Response:"
```

with `EXAMPLE` substituted by our addition example. For instance, if we prompt the model with

```
"Below is an instruction that describes a task."
"Write a response that appropriately completes the request.\n\n"
"### Instruction: \n5542 + 2067 = \n\n### Response:"
```

the response we got is `7519` (we omitted BOS and EOS tokens). In the majority of the responses the response was just an integer, but sometimes it would repeat the sum and give the answer as $a + b = c$. We removed all those additional tokens so that we would be left with just the model's answer for the sum. We used that to calculate the accuracy. After the model generated the responses we looked at the attention patterns and searched for the *staircase* pattern we have been seeing in the smaller models. We did this search for every generated token, so at step 0 in the generation we get the attention pattern of the input prompt, which we search and then at subsequent steps in

---

[6]It turns out that for models with more than one layer, the biases are irrelevant and can be set to zero without affecting the model's performance. For one-layer models the biases are important, which is curious.

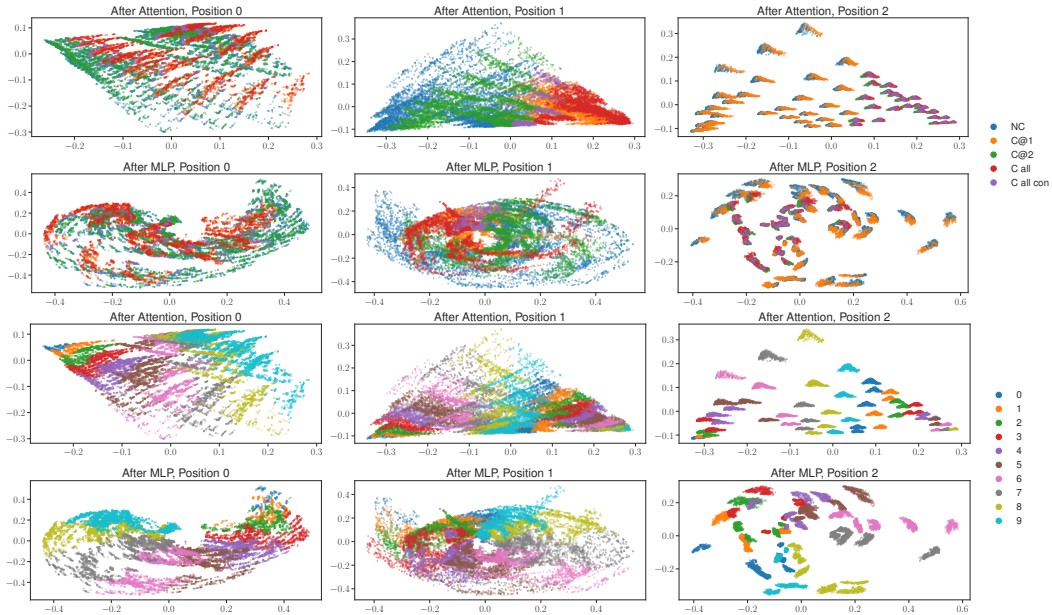

Figure 7: PCA analysis for the outputs of the attention and MLP blocks. Columns are positions $0$, $1$ and $2$ of the outcome. Rows $1, 2$: after attention, MLP (resp.) labelled according to the five tasks, `NC` (blue), `C@1` (orange), `C@2` (green), `C all.` (red) and `C all con.` (purple). Rows $3, 4$: after attention, MLP (resp.) labelled according to the digit in the answer at each position.

the generation we just focus on the newly generated row in the attention map. We then record the frequency of each head and consider the 8 most frequent heads. We then study ablating pairs of heads in this list, starting with the most frequent pair and if that did not affect accuracy we considered other pairs in this list of 8 most frequent heads.

For the ablation study, we modified the huggingface implementation of the LLaMA models so that they gave not only the output of each layer, but also after the attention. Furthermore, we added a routine to ablate a specific head.

Here we focussed one one single way of prompting the model to do addition, but there are perhaps better ways of doing it that get higher accuracies. It would be interesting to explore that, but might make interpretability more complicated.

### A.4 LLEMMA 7B AND ZEPHYR 7B

We prompted Llemma 7B Azerbayev et al. (2023) and Zephyr 7B Azerbayev et al. (2023) using just `a + b =` and using the same sums as for Alpaca. We can again use the huggingface implementation of the LLaMA models to do the ablation tests.

## B A JOURNEY OF THE HIDDEN REPRESENTATIONS IN ONE-LAYER MODELS

In this appendix we elaborate a bit more on the residual stream of one-layer models for the model discussed in the main text with $s = 0.3$ and $\lambda = 0.2$. We again look at $20k$ examples from the entire dataset. We then perform a principal component analysis to isolate the important directions after the attention and MLP blocks. For the $20k$ examples we distinguish between the five cases mentioned in Sec. 2 and project the data to the two leading principal axes. The results are shown in Fig. 7.

There are a few interesting things we can note from this figure:

1. In the first row (i.e. after the attention) we see examples are roughly grouped according to whether the sum at a given position was $\geq 10$ or not. At position 0 they are all grouped together, because it is always less than $10$, at position 1 the examples with no carrying

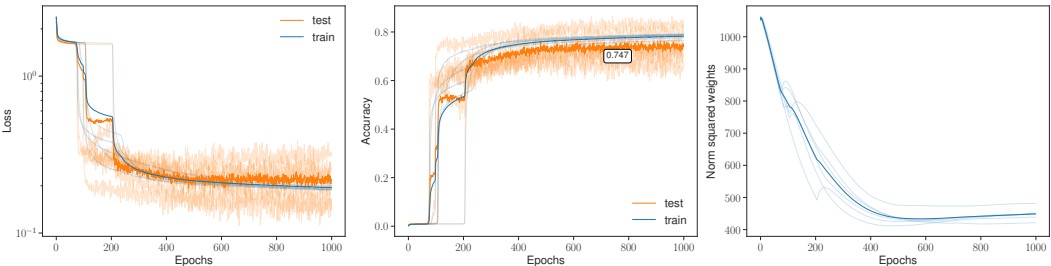

Figure 8: Loss, accuracy and norm of the weights as a function of epochs for both the training and test data. Here the train/test split is $s = 0.3$ and $\lambda = 0.2$. The accuracy is computed as the minimum over all three positions in the answer and we plotted six runs and their average (bold). The final average test accuracy is shown boxed.

     over and carrying over at position 2 are grouped, since only for those the sum is $\geq 10$ and for non-carrying over and carrying at position 1 the sum is always $< 10$ and so are also grouped. Similarly for position 2.

2. Within each of the groups in the first two rows, the examples are grouped according to what their outcome is going to be. However, for positions 0 and 1 there is a big spread with a lot of overlap. This indicates that when examples are different, the model does not localize them according to some underlying structure but just gives them a different location in the residual stream. For position 2 there are more localized pockets. This is very different from the two-layer models discussed in Sec. 4 as there we see a clear division in the residual stream.

3. Despite there being this big spread, the datapoints corresponding to a particular outcome at a certain position do follow the ordering of the digits 0 through 9.

4. Examples are never grouped according to whether a carried one needs to be added or not. So it does not implement the carrying over algorithm. This is again very different from the two-layer models, which do 'realize' this.

The first three points are actually rather remarkable, since this information is not supplied while training and seem to indicate that the model is ordering the data in an rather interesting way. Especially the fact it learned to order of the digits is something we did not expect in such a small model to occur. The same behaviour we saw in the other runs with these hyperparameters. For other weight decay or train/test split we see the same structures, but if we turn off weight decay however, this structure seems to be lost.

The PCA in Fig. 7 shows some spiral like behaviour, which usually indicates one needs a non-linear version of PCA, which might be fruitful to study.

## C    MORE MODELS

Here we discuss some more results on models trained with different train/test split and weight decay. All models discussed here have $d_{\mathrm{model}} = 128$ and $d_{\mathrm{ff}} = 128$ unless stated otherwise. One can see that with our digit representation we did not see any grokking.

### C.1    ONE LAYER

As we mentioned, the one-layer models are not necessarily worthwhile for studying their performance, but rather they exhibit an interesting transition in their QK circuit.

$s = 0.3$ **and** $\lambda = 0.2$   : In Fig. 8 we plotted the loss, accuracy and norm-squared of the weights for all the runs we performed for the hyperparameters discussed in the main text. Again, each run has a transition in its QK-circuit.

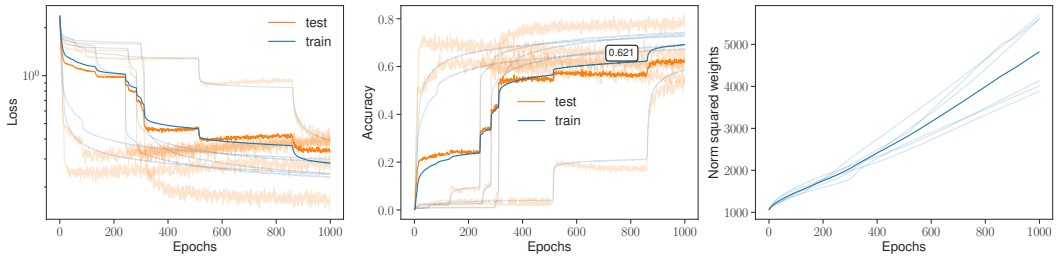

Figure 9: Loss, accuracy and norm-squared of weights as a function of epochs for $s = 0.3$ and *no* weight decay. Thick lines are averages over six runs.

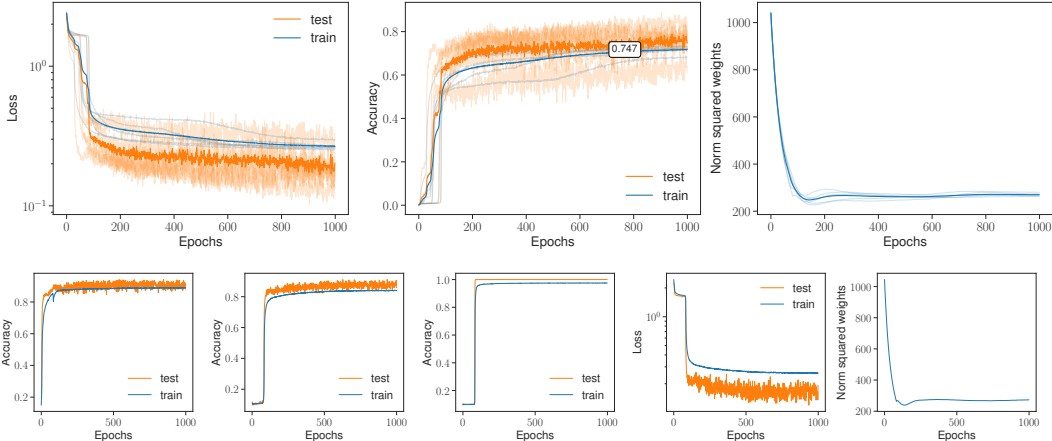

Figure 10: **Top**: Loss, accuracy and norm of the weights as a function of epochs for both the training and test data. Here the train/test split is $s = 0.2$ and $\lambda = 1.0$. The accuracy is correctness of the answer and we plotted six runs and their average (bold). The final average test accuracy is shown boxed. **Bottom:** Loss, accuracy and norm of the weights for a specific run. The first three plots are accuracies for each position.

$s = 0.3$ **and** $\lambda = 0$ : For these six runs we also turned off weight decay, but kept the original $s$. We again see a large variance. The runs learned to some reasonable accuracy, but for some the test loss started increasing after some epochs. Longer runs would conclude what happens when the weights become too big. See Fig. 9. It again suggests that weight decay is important to get a consistent and stable output regardless of the initialization.

$s = 0.2$ **and** $\lambda = 1.0$ : As can be seen in Fig. 10 all six runs exhibit the transition.

## C.2 TWO LAYER

$s = 0.3$ **and** $\lambda = 0.2$ : The learning dynamics for the six runs can be found in Fig. 11, where we also plotted the per-token accuracy for the run discussed in Sec. 4.

The PCA done in the main text was for a particular run. For the other five runs we also saw the same structure: layer 0 groups according to whether sum is $< 10$ or not and layer 1 determines whether a sum needs a carried one.

$s = 0.3$ **and** $\lambda = 0$ : For these six runs we turned off weight decay. Often weight decay can drive the model into adopting a certain algorithm. For instance see Nanda et al. (2023) (although, here it was also important to have weight decay to drive grokking). In our case the learning dynamics did not show any significant effect from the absence of weight decay; the model did again reach perfect accuracy and the loss decayed. See Fig. 12.

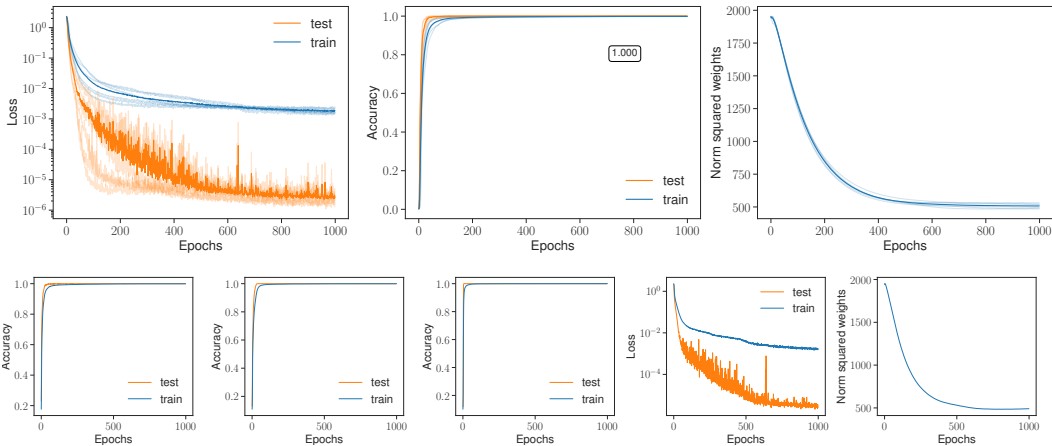

Figure 11: Learning dynamics of the two-layer model with $s = 0.3$ and $\lambda = 0.2$. **Top**: Loss, accuracy and norm-squared of all the weights as a function of epochs. Six runs are plotted with their average in bold. **Bottom**: Per token accuracy, loss and norm-squared of weights of one of the six runs.

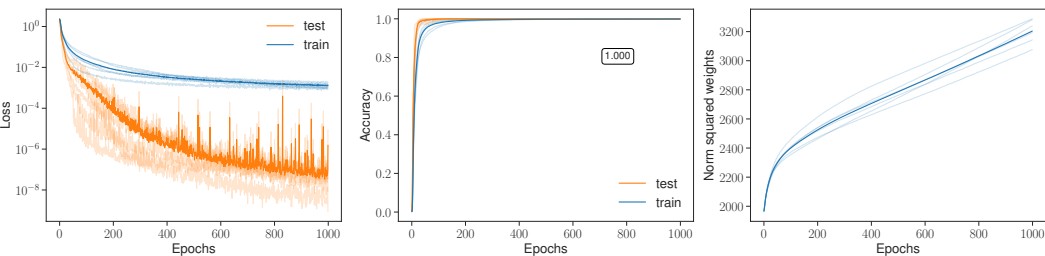

Figure 12: Loss, Accuracy and norm-squared of weights for six runs with $s = 0.3$ and *no* weight decay. Solid lines are averages over six runs.

One reason might be that we still have dropout turned on and so some regularization is present. Nevertheless, the models still seem to develop the carrying over algorithm as described in other cases. However, ablating the full final MLP only worked for one run, the others required dissecting, but that turned out to work successfully also.

$s = 0.1$ **and** $\lambda = 0.2$ : Very similar learning dynamics and perfect accuracy was obtained, but weights did not saturated after 1000 epochs. For some runs the ablation of the MLP is too much and we used the criterion discussed in the main text to identify a set of neurons responsible for carrying over the one. For the PCA analysis performed in the main text, we see most of the models have the same structure as in the main text, but for one run we saw a clear difference in the second layer. In particular, it did *not* use the second attention block to determine sums that needed a carried one or not. This is curious, but when we then study the attention pattern for that particular run, we see the model has already transferred the information about the sum of a previous position to the current one in the *first layer*'s attention, instead of the final layer. Despite this, the MLP still carries the one (after dissecting it properly) and so for these hyperparameters the carrying over algorithm is again implemented.

$s = 0.3$, $\lambda = 0.2$ **and** $d_{\text{ff}} = 1024$ : The dynamics for this model look rather similar to the ones we encountered in the main text. There are some differences though. For most runs, ablating the entire final MLP affects accuracy throughout, not just carrying of the one. However, one can again identify a specific set of $\sim 650$ neurons that are relevant for carrying of the one. The attention patterns and PCA as discussed in the main text are again very similar, i.e. there is the same separation between types of examples. Thus again the carrying over algorithm is implemented.

Table 3: Non-carrying over versus carrying over at specific positions after ablating the final MLP block of the three layer model with $s = 0.3$ and $\lambda = 0.2$. The non-carrying over tasks are not affected much, but the model significantly reduced in its ability for carrying the one. It is again either correct or off by one (forgot to add a carried one or added one where it shouldn't). It is interesting to note that it does know how to deal with the final task at the second position.

| Task | acc. pos. 7 | acc. pos. 8 | acc. pos. 9 |
|---|---|---|---|
| NC | **0.90** | **0.93** | **0.95** |
| Corrected NC | 0.10 | 0.07 | 0.05 |
| C@1 | 0.19 | **0.87** | **0.95** |
| Corrected C@1 | **0.81** | 0.13 | 0.05 |
| C@2 | **0.90** | 0.15 | **0.88** |
| Corrected C@2 | 0.10 | **0.85** | 0.12 |
| C all | 0.18 | 0.21 | **0.88** |
| Corrected C all | **0.82** | **0.79** | 0.12 |
| C all con. | 0.19 | **0.99** | **0.89** |
| Corrected C all con. | **0.81** | 0.01 | 0.11 |

## C.3 THREE LAYER GENERALIZATION

$s = 0.3$ **and** $\lambda = 0.2$ : The learning dynamics is very similar to the two-layer models. Starting from the first layer, we notice the model again found the relevant attention patterns and in the PCA for the first layer we see exactly the same structure as in the two-layer model: after the first layer the model has grouped together examples depending on whether the sums at each position were $< 10$ or not. Here also, a 9 is considered a special case. For the second layer it is already sorting the examples according to which one needs a carried one. In the third layer the model has separated the examples in exactly the way we also encountered in the two-layer model.

This suggests that determining whether a carried one needs to be added or not is now distributed not over just one head, but perhaps over everything before the final MLP and after the first layer.

Looking at the final MLP, we find that ablation it entirely results in a drastic reduction in the ability to carry the one, see Tab. 3

For identifying what heads are important in determining which sums need a carried one or not, we did not find the same structure as we found in Tab. 1. That is, after ablating each of the heads in the last two layers, we did not find a significant effect on the accuracy. It seems plausible that the model now uses heads in both the second and third layer to accomplish this. In fact, if we ablate the entire second layer attention and one of the two final attention heads, there are some runs that have exactly the same structure as discussed in the main text: high variance on non-carry sums, but very sure about where to add a carried one whenever it should. Again we saw the division was exactly between correct and off by one.

At any rate, we think it is fair to say that a large part of the carrying over algorithm are implemented, but due to the additional layer, the attention heads of both the second and third layer seem to work together in an intriguing way.

## C.4 MORE DISSECTIONS OF THE MLP

Let us report on a few more dissection examples:

$s = 0.1$, $\lambda = 0.2$. There we found that only some runs could handle a full ablation of the MLP and some did not. For the latter runs, we ablated part of the neurons using the prescription just outlined (which resulted in a set of around $\sim 64$ different neurons) and found that non-carrying over is not affected *at all*, but carrying over went down to $0.57$ (the other $0.43$ it forgot to add a one).

$s = 0.2$ **and** $\lambda = 1.0$, $d_{\text{ff}} = 512$ The results after ablating $\sim 282$ neurons in the final MLP are shown in Tab. 4. Besides the ablation study we also performed the SVD analysis discussed in the main text to identify a set of carrying over neurons. We did an SVD on the pre-activation weights

Table 4: Non-carrying over versus carrying over at specific positions in one model with a specific part of the final MLP block ablated. The non-carrying over tasks are not affected, while the model significantly reduced in its ability for carrying the one. Notice that position 9 is unaffected as these are all non-carrying sums. Here $d_{\text{ff}} = 512$, $s = 0.2$ and $\lambda = 1.0$

| Task | acc. pos. 7 | acc. pos. 8 | acc. pos. 9 |
|------|-------------|-------------|-------------|
| NC | **1.0** | **1.0** | **1.0** |
| Corrected NC | 0.0 | 0.0 | 0.0 |
| C@1 | 0.05 | **1.0** | **1.0** |
| Corrected C@1 | **0.94** | 0.0 | 0.0 |
| C@2 | **1.0** | 0.15 | **1.0** |
| Corrected C@2 | 0.0 | **0.84** | 0.0 |
| C all | 0.19 | 0.23 | **1.0** |
| Corrected C all | **0.78** | **0.67** | 0.0 |
| C all con. | 0.14 | 0.03 | **1.0** |
| Corrected C all con. | **0.85** | **0.97** | 0.0 |

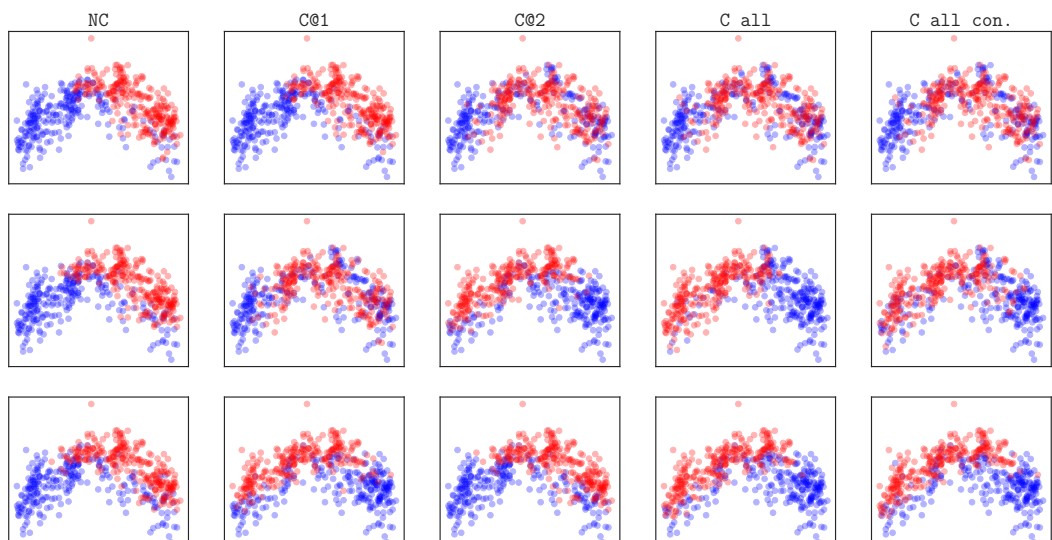

Figure 13: SVD analysis for the pre-activation weights of the last layer in a model with $s = 0.2$ and $\lambda = 1.0$. We plot the leading axis as the $x$-axis and the sub-leading one as the $y$-axis. We see the leading one represents a *carrying over* feature. Red means active, blue is non-active. The five columns represent the five tasks and the three rows the positions in the outcome. Again we used 20k examples to generate these plots and plotted the 256 most active neurons. For clarity we did not plot any axis labels/ticks.

and then checked which neurons got activated with a particular task and (outcome) position. We plotted the 256 most activated neurons for each case and this resulted in Fig. 13.

From this figure we clearly see that whenever a carried one is necessary the right side (larger value along leading axis) of the neurons is mostly activated whereas for non-carry sums the left hand side is more active as is clear from the first column. For the second column the final position (third row) the left is more active as it should for C@1 as is the second position for the C@2 column. For the other columns one sees a similar pattern.

$s = 0.3$ **and** $\lambda = 0.2$    The SVD analysis for the 64 most active neurons is shown in Fig. 14. Again there is a clear set of neurons that are activated when a carried one needs to be added.

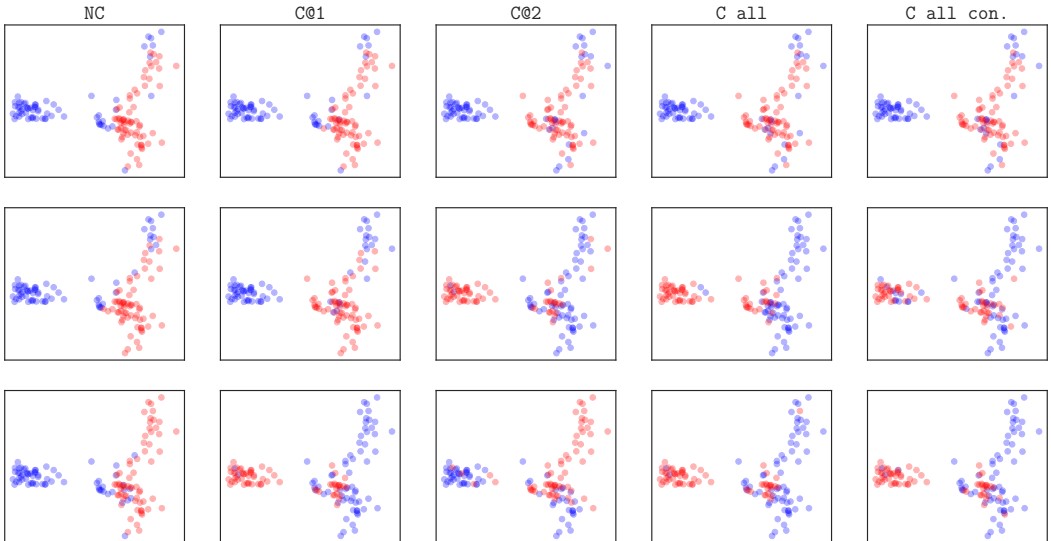

Figure 14: SVD analysis for the pre-activation weights of the last layer in a model with $s = 0.3$ and $\lambda = 0.2$. We plot the leading axis as the $x$-axis and the sub-leading one as the $y$-axis. We see the leading one represents a *carrying over* feature. Red means active, blue is non-active. The five columns represent the five tasks and the three rows the positions in the outcome. It is clear that the left blob is active when a carried one is needed. Again we used $20k$ examples to generate these plots and plotted the 64 most active neurons. For clarity we did not plot any axis labels/ticks.

# D  MORE DETAILS ON GENERATIVE MODELS

We used the same architecture of the models in the main text for the generative models. The decoder-only models now have a causal mask and we train on the shifted logits as usual. For this we altered the three digit addition dataset by concatenating the inputs (or prompt) with the target and used an = sign as the end-of-sequence token. We used the same training hyperparameters and model size as in the main text.

We find the same implementation as for encoder-only models, but with some detailed differences in its implementation due to the causal nature. These differences can be seen in the attention pattern. While there is still an staircase pattern, the models can only move information causally and so the staircases appear at specific places, see Fig. 15. In layer 0 there is some attention tokens pay towards themselves, but more importantly information from the first integer is stored in the second and added using the skip connections. In layer 1 we see how the model transfers that data together with sums at earlier positions to compute the output at the current position. Here we see how this happens generatively. The staircases in layer 1 for the generative step (final four rows) are shifted as the final token predicts the next one. We also see how again there is one head (1:1) that transfers the carrying over data from the previous sum, whereas head 1:0 deals with the current sum.

Next, if we look at the outputs of the attention and MLP at each layer, we see a very similar pattern as we encountered for encoder-only models. The resulting PCA is plotted in Fig. 16. The clustering of examples is very similar to the encoder-only models. For layer 0 a clear distinction is being made between whether a sum is $< 10$ or not. This is also true for the naive sum (no carried one and modulo 10) of the digits at the same position of each integer. In particular for pos. 4 we see only a grouping according to the naive sum but no grouping for the task as all tasks at that position have a sum $< 10$. At pos. 5 we see examples C@1 and C all grouped together, C all con. grouped together (compare with the second row to see that this can only have a 9 as outcome) and C@2 and NC are grouped together. This is what one expects if the grouping is according to the value of the naive sum. For pos. 6 we see something similar. In layer 1 the situation is a little less clear, but again the attention grouped examples according to whether a carried one is needed or not and the final MLP groups them according to their target value. In fact there seems to be a rough pentagon

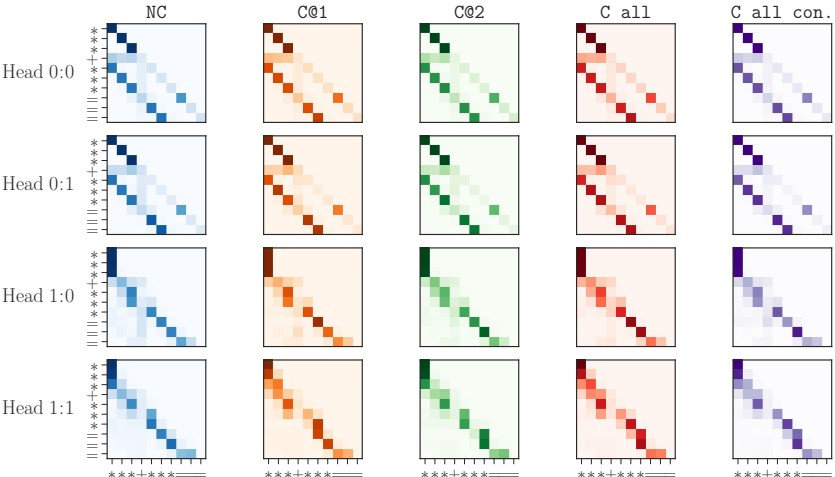

Figure 15: Attention pattern for a decoder-only model with $s = 0.3$ and $\lambda = 0.2$ per task (see Sec. 2. The staircase patterns are still there and the model again moves information from the first integer to the second in the first layer and processing it to the output in the second layer. Head 1:1 acts as a *decision* head as it supplies the additional information needed from previous sums.

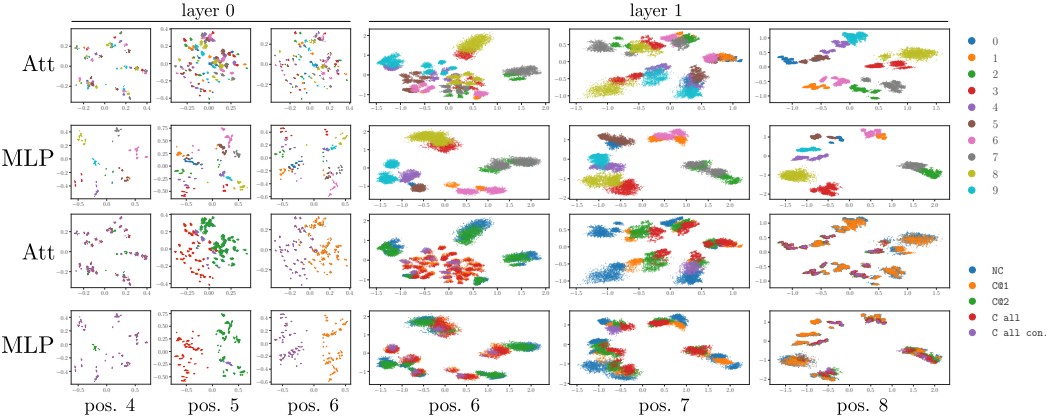

Figure 16: PCA analysis for a generative model. We plotted positions 4-6 for layer 0 and 6-8 for the second layer. The data in the first two rows is labelled according to their target value at that position (layer 0 does not take a carried one into account, but layer 1 does) and for the last two rows we labelled according to the five tasks. We see that the model again, after the first MLP has grouped examples according to their (naive) target value and whether the sum is $< 10$ or not. For layer 1 the output of the attention is grouped according to whether a carried one is needed or not, but it is not as clear is for the encoder-only models. The output of the MLP is again a set of localized spots associated with the target value, similar to encoder-only models.

structure, which we encountered in other models too and is something that we will discuss further in E.

An ablation study of the models' final attention heads also shows that head 1:1 is responsible for determining whether a sum needs a carried one or not. If ablated a carried one is always added and again the accuracy is divided between being correct or off by one, just as in the main text's examples. If instead we ablate head 1:0 (removing information about the current sum) all carrying over sums see a dramatic drop in accuracy down to $0.34$.

Finally, the MLP in the second layer is again responsible for the algebraic step of adding the carried one. To show this, we can do a ablation study of the entire MLP or dissect it using the procedure

Table 5: Accuracy after ablating the final MLP and dissecting it for the five tasks in case of a generative model. The 'corrected' accuracies for non-carry sums are obtained by manually subtracting one from each position of the output of the model and comparing that with the target. In this way we can see how many of the non-carry sums got an incorrect carried one. For carrying over sums we did the same, except we added a one so as to see for what example the model forgot to add a carried one. Averages over $100k$ examples.

| Task | Ablating | | | Dissecting | | |
|---|---|---|---|---|---|---|
| | pos. 7 | pos. 8 | pos. 9 | pos. 7 | pos. 8 | pos. 9 |
| NC | **0.99** | **0.92** | **0.96** | **1.0** | **1.0** | **1.0** |
| Corrected NC | 0.01 | 0.08 | 0.05 | 0.0 | 0.0 | 0.0 |
| C@1 | 0.30 | **0.98** | **0.94** | 0.27 | **1.0** | **1.0** |
| Corrected C@1 | **0.70** | 0.02 | 0.05 | **0.73** | 0.0 | 0.0 |
| C@2 | **0.99** | 0.28 | **0.99** | **1.0** | 0.16 | **1.0** |
| Corrected C@2 | 0.01 | **0.72** | 0.0 | 0.0 | **0.81** | 0.0 |
| C all | 0.29 | 0.11 | **0.97** | 0.27 | 0.05 | **1.0** |
| Corrected C all | **0.71** | **0.87** | 0.01 | **0.73** | **0.93** | 0.0 |
| C all con. | 0.25 | 0.0 | **0.97** | 0.27 | 0.0 | **1.0** |
| Corrected C all con. | **0.75** | **1.0** | 0.02 | **0.73** | **1.0** | 0.0 |

described in the main text. The result of both of these experiments can be seen in Tab. 5. For the dissection we found around 92 neurons to be relevant for carrying the one.

All in all, we can conclude the carrying over algorithm is implemented.

## E   PENTAGON STRUCTURES IN THE RESIDUAL STREAM

In the main text we saw already some interesting examples of structures in the residual stream, but they were mostly aligned with the SVD directions. Here we want to discuss an example that was not aligned with those directions as we already alluded to in the main text. Specifically we encountered situations where the data was arranged in a periodic way, utilizing not only the ordering of the digits but also grouping them a fixed distance. For instance, see Fig. 17.

In this figure we see that at position $8$ (second position in the outcome) for the tasks that do not require a carried one at that position, there is a pentagon structure where the vertices are pairs of integers with distance 5, i.e. $\{0, 5\}, \{1, 6\}, \{2, 7\}, \{3, 8\}$ and $\{4, 9\}$ and they are ordered cyclically. The model has thus formed a representation of the digits in terms of these five pairs.

More mathematically this means the model forms a real two dimensional representation of $\mathbf{Z}_5$. For the tasks with a carried one the model realizes a different representation, obtained by rotating by one vertex. This makes sense of course, but does raise the question, how it knows the difference between whether a carried one is needed or not. It turns out that this information is captured in the subsubleading direction as can be seen in Fig. 17.

This subsubleading direction is arranged not only according to whether a carried one is needed or not, but as can be seen it also separates numbers that are $< 5$ from those that are $\geq 5$. This makes the representation in the leading two dimensions projective as when we traverse the pentagon once, we jump in the subsubleading direction.

## F   4 DIGIT ADDITION

In this appendix we analyse a specific run for $4$ digit addition. We use the same two-layer model as in the main text: $s = 0.3$ and $\lambda = 0.2$, learning rate $1.4 \times 10^{-4}$, but changed the batch size to 8192. The entire dataset consists of $\sim 5 \times 10^7$ examples, so we are way off in the underparametrized regime. Yet, our model reaches perfect accuracy and it does so by implementing the carrying over algorithm in the way we discussed in the main text. Let us discuss this in some more detail.

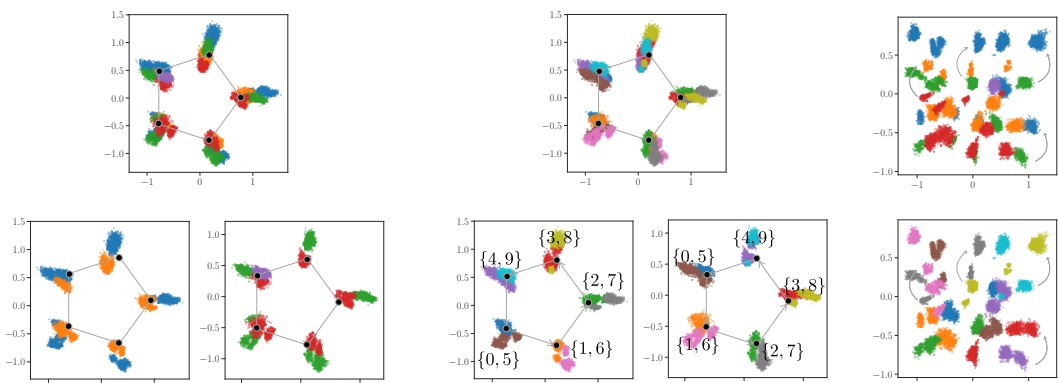

Figure 17: PCA of position 8 (second position of the outcome) after the attention in the second layer for a model with $n = 2$, $s = 0.3$ and $\lambda = 0.2$. **Left:** Labelling according to the five tasks. We see a clear pentagon structure, and we have exposed the tasks that require carrying over or not below it on the right and left respectively. **Middle:** Same plot but now labelled according to the digit in the target's second position. This reveals that the vertices of the pentagon are pairs of integers separated by 5 and are cyclically ordered (counter-clockwise). The separation by tasks as in the left panel shows that the two pentagons are rotated relative to each other as it should. **Right:** This shows the subleading ($x$-axis) and subsubleading ($y$-axis) principal axes and we see the subsubleading direction not only separates data depending on whether a carried one is needed or not, but also depending on whether the integer is $< 5$ or $\geq 5$. With the arrows we indicated that the tasks and integer $> 5$ versus $\geq 5$ is separated by a roughly fixed translation in the $y$ direction.

For three digit addition we separated the examples in five different groups according to their carrying over pattern. Now we have thirteen such classes:

1. NC: $a_i + b_i < 10$

2-4. C@j: $a_j + b_j \geq 10$, $a_i + b_i < 10 \; \forall \; i \neq j$

5. C@12: $a_{1,2} + b_{1,2} \geq 10$, $a_3 + b_3 < 10$

6. C@13: $a_{1,3} + b_{1,3} \geq 10$, $a_2 + b_2 < 9$

7. C@23: $a_{2,3} + b_{2,3} \geq 10$, $a_1 + b_1 < 9$

8. C@12 con.: $a_1 + b_1 = 9$, $a_2 + b_2 \geq 10$, $a_3 + b_3 < 10$

9. C@23 con.: $a_2 + b_2 = 9$, $a_3 + b_3 \geq 10$, $a_1 + b_1 < 9$

10. C@12p con.: $a_1 + b_1 = 9$, $a_2 + b_2 \geq 10$, $a_3 + b_3 \geq 10$

11. C@23p con.: $a_2 + b_2 = 9$, $a_3 + b_3 \geq 10$, $a_1 + b_1 \geq 10$

12. C@all con.: $a_{1,2} + b_{1,2} = 9$, $a_3 + b_3 \geq 10$

13. C@all: $a_i + b_i \geq 10 \; \forall \; i > 0$

**Attention** Let us start with the attention patterns, see Fig. 18 (zoom-in required). We see that the attention patterns again have the staircase pattern where digits pay attention to the relevant digit in the other integer. The two heads in the first layer are mostly similar, but in the second layer we see the second head (head 1:1) is an *decision* head. It carries the information from the previous position to the current one. For each task we see most probability is on the digit from the previous position relevant for that task. We can confirm the importance of this head with an ablation study. Again when we remove this *decision* head the accuracy is exactly divided between correct or off by one, just as before.

**MLP** Moving on to the final MLP block we see exactly the same function as before for three digit addition. In fact, for this run, we can ablate the entire MLP and *all* examples that require carrying of the one now have accuracy equals 0 at that position, but equals 1 if we manually add a carried one!

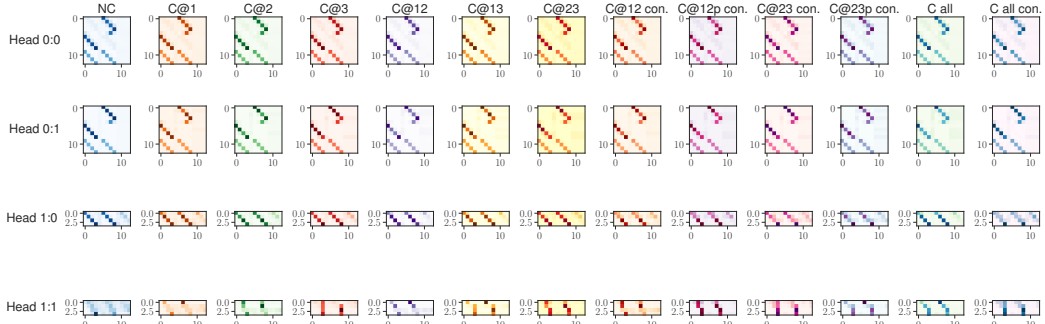

Figure 18: Attention pattern for each head and layer. The staircase pattern in the first layer is either indicating attention between digits at the same position of each integer (first 9 rows) or the outcome at a certain position receives attention from two digits at the relevant positions (last four rows). For the second layer attention we only plotted the last three rows and we see head 1:1 transfers information from the sums at a previous position to the current one. It is an example of a *decision* head.

We can also see what neurons are responsible for the carrying of the one and found a set of $\sim 94$ of such neurons.

Thus the conclusions we reached for three digits *generalizes* to four digit addition.

### F.1 A JOURNEY OF HIDDEN REPRESENTATIONS

We will be a bit briefer here as compared to the discussion in the main text. The squashing we mentioned is still present for the experiment with a fixed outcome. For the second layer we obtain, for each position, $0.24$, $0.22$, $0.16$ and $0.21$ respectively. As for the PCA analysis, which gets a little bit cumbersome, the same patterns persist: The first layer determines whether sums are $< 10$ or $\geq 10$, whereas the second layer determines which sums need a carried one or not. See Fig. 19

Thus we conclude that the the carrying over algorithm is again implemented.

## G MORE RESULTS ON ALPACA 7B

To get a more detailed understanding of how Alpaca is doing integer addition, requires a much better accuracy than the one we obtained. Luckily, the accuracy per position, is actually reasonably high for the firstly generated position ($0.93 \pm 0.25$). Looking at the first two principal axis in the residual stream after the attention and MLP block at each layer we find that at each newly generated position Alpaca manipulates that data such that at the final layer's output, the leading axis corresponds to *whether a carried one was necessary at the firstly generated position or not*. This is similar to what we saw in the smaller models after the final layer's attention. Besides separation in tasks, which seemed to be happening throughout the network, we also observed the leading axis being used to order the data according to the target output. This, however, only happened in the middle of the network, suggesting perhaps that the model performs some calculation there, but moving the result off to another dimension in the residual stream and recalling it at the end.

## H LENGTH GENERALISATION

We studied models trained on three digits and tested them on 6 digits sums. This requires inputs to the models to have some additional padded positions, which will pad with 0s on the left (using other non-numerical padding tokens seems to not have an impact)[7]. We train for 1000 epochs and

---

[7]Just giving the models in Sec. 4 the six digit sums fails dramatically.

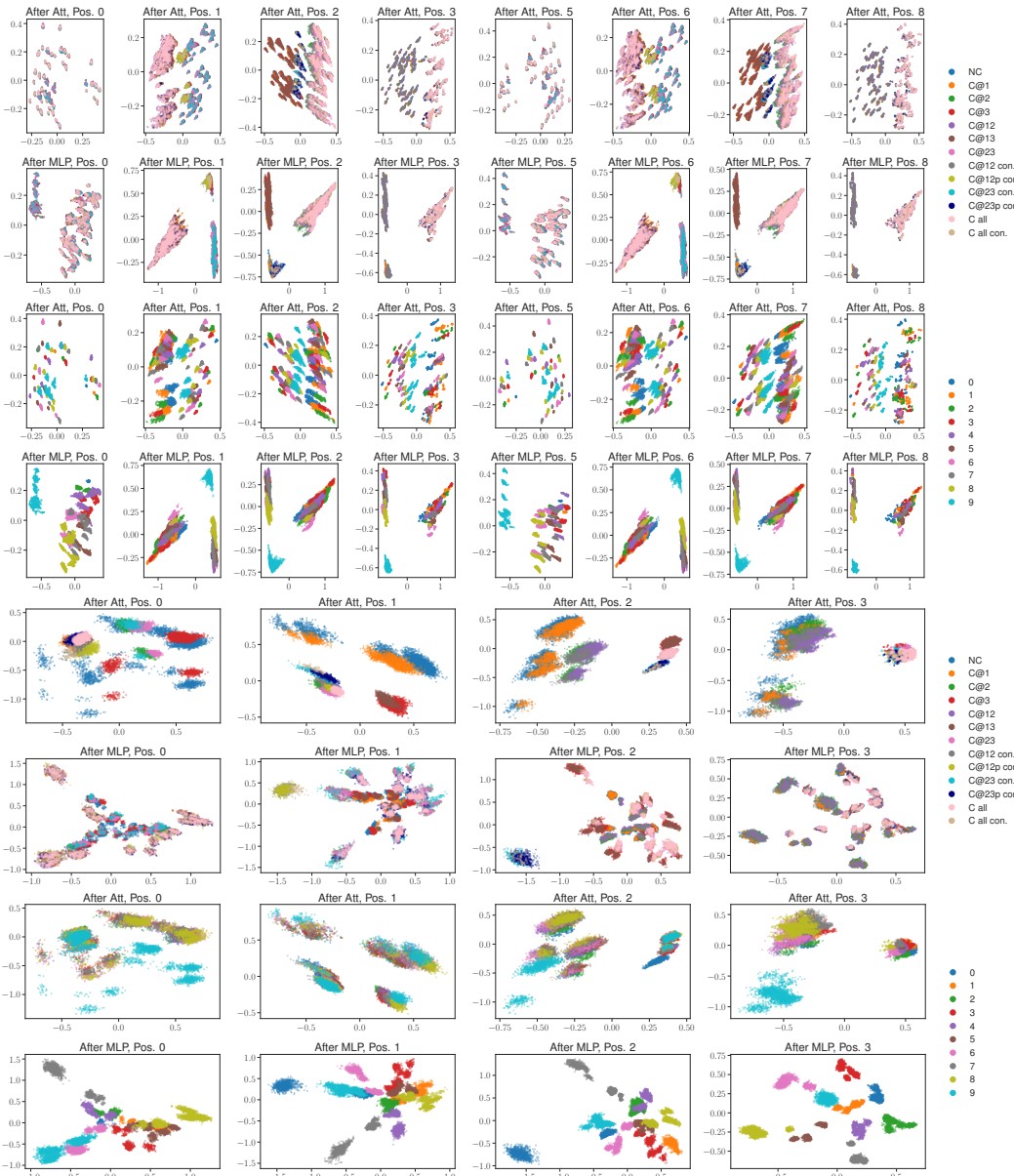

Figure 19: PCA analysis for the outputs of the attention and MLP blocks in each layer for the two leading principal axes. Rows $1, 2, 5, 6$ are colored according to the 11 tasks. Rows $3, 4$ are labelled according to the sum (ignoring any carried one) at that position. Rows $7, 8$ are labelled according to the answer at that position. We see the first layer determines whether sum $< 10$ or $\geq 10$ and groups those examples (separating also the $9$ as a special case). The second layer instead groups examples according to whether they need a carried one or not. Notice that position 3 never needs a carried one and so the examples are grouped in the same way as in the first layer. Furthermore, after the second Attention at position $0$, examples that need a carried one are grouped together, but they are not as cleanly separated from the rest.

use the same hyperparameters as before. The resulting test set accuracy per position of a model with $s = 0.3$ and $\lambda = 0.2$ is shown in Fig. 20 in blue.

We see that around epoch 500 the model starts to drop in accuracy for first position. Consequently, the total accuracy (correctness of the output) also drops and goes from around 0.20 at epoch 500

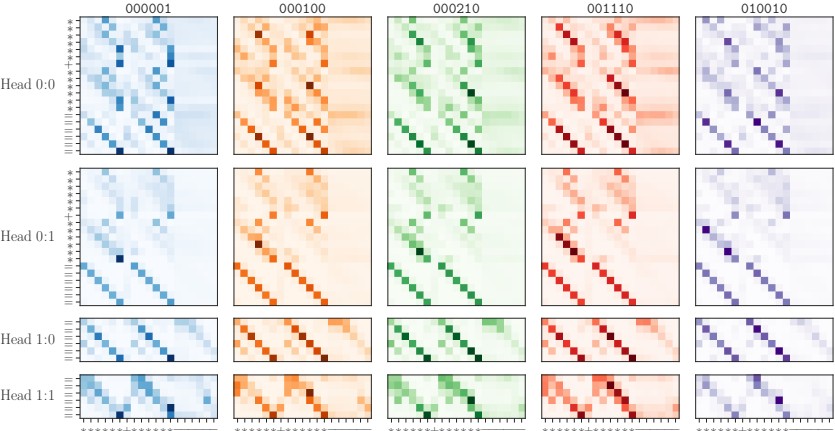

Figure 20: Test set accuracy per position on 6 digit addition as a function of epochs, $s = 0.3$ and $\lambda = 0.2$, with (orange) and without (blue) priming.

Figure 21: Attention pattern for 5 out of the 89 different cases for 6 digit addition. The cases are labelled using a trinary system, where a 0 means $< 9$ at this position, 1 means $\geq 10$ and 2 means equal to 9 (if previous sum was $\geq 10$, i.e. this is for carried ones that propagate further than just one position.). Apart from the clear staircase patterns, we also see that, for instance, head 1:1 transfers information from the previous sum to the current one. E.g. looking at case `001110`, we see position 1, 2 and 3 in the final output positions get attention from the sums at positions 2, 3 and 4 as we would expect. For other cases we see something similar. It is also interesting to see that some information of the previous sum is also transferred already by head 0:0, so it is doing the addition at the current position and collection of information from previous sums in parallel.

to 0.02 at epoch 1000. We are not sure why this is the case and seems to happen for different hyperparameters and initializations.

Let us analyze this model at epoch 500 a bit more. For instance, consider the attention patterns in Fig. 21. We see that despite the rather low accuracy, the model does know what the correct attention is, i.e. the staircase attention we saw before. Furthermore, we have displayed the attention for a few cases (there are 89 in total) and from those we also see that there is again a head that transfers information from the previous sum to the current one.

What is interesting is that when training goes on beyond 500 epochs, the model seems to destroy some of these attention patterns for the OOD positions, which might trigger the bad accuracy at those positions.

The attention patterns in layer 0 also suggest that the residual stream at the output positions (the equal signs) has a nice structure as there the sums of the current are being calculated. As we saw before, the model structured the two leading principal directions so that cases for which the sum is $\geq 10$ or $< 10$ are separated. This is indeed what we see in the first layer, but as we get to the OOD positions (first three columns) the separation is not as good anymore, see the first two rows in Fig. 22. For layer 1 we see the separation is now according to whether a carried one is needed or not, just as for the analysis in Sec. 4.

One thing that we have not seen in the models trained in this way, is the MLP being responsible for the carrying of the one. Given the evolution discussion in Sec. 5, this might be because the MLP is under construction still. Indeed, training the model further, we see that after a total of 2500 epochs,

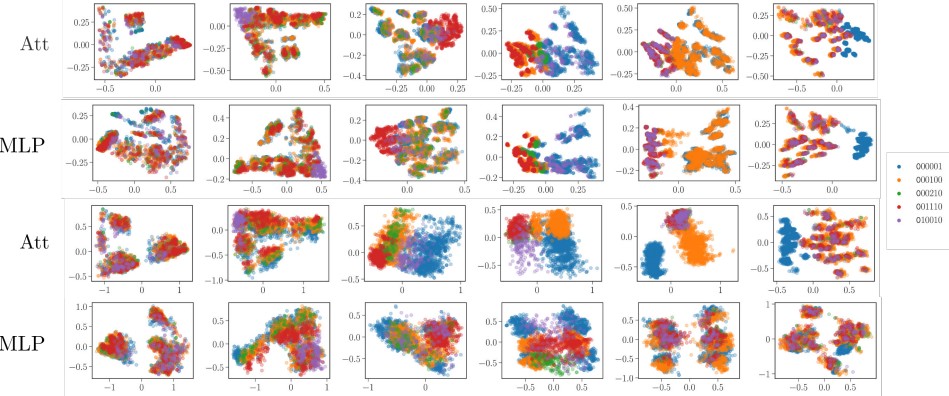

Figure 22: PCA of the residual stream after the Attention and MLP of the first and second layer, respectively. We labelled the data according to the five tasks discussed in Fig. 21. The six columns are the six output positions (last six positiosn). We used 20$k$ examples.

the MLP is much more clearly responsible for the carrying of the one, albeit for the ID positions. For the OOD positions, the model has a pretty bad accuracy at those positions, so there is no hope of establishing a clear function of the MLP.

To summarize we find the following: i) A length 6 staircase in the attention patterns in both layers. ii) There is a particular head that transfers the information of the previous sum to the current one iii) The leading principal axis of the PCA of the hidden states is organised in the same way as discussed in Fig. 4 for the ID (in-distribution) positions 3, 4 and 5, but for the OOD (out-of-distribution) positions 0, 1 and 2, this is not the case generally. Some cases (of the 89 in total) are separated according to the same principals, but it is not as clear as for ID positions. Also, the MLP (even after dissecting) is not a clear performer of the carrying of the one. These two observations might be the source of the poor performance and it would be interesting to study this further. Furthermore, the decay of the OOD accuracy seems to be also driven by a deterioration of the attention patterns.

What do we learn from this behaviour? At 500 epochs, the model seems to have most of the ingredients put in place to construct an efficient implementation of the carrying over algorithm. For the three ID positions this implementation improves, but deteriorates for the other three positions and part of the challenge is to prevent this from happening. This means in particular understanding how the attention patterns can be stabilized.

### H.1 PRIMING

One way to do so, as discussed in Jelassi et al. (2023), is to prime the model with a tiny set of examples. For instance, we can prime our 3 digit dataset, consisting of 150150 examples (remember we take a split $s = 0.3$) with just 100 (random) six digit examples, i.e. 0.07% priming. Again we train for 1000 epochs with the same hyperparameters. The test accuracy per position on six digit sums of such a (primed) model is shown in Fig. 20. It is clear that even a small amount of examples indeed has tremendous effect on the model's ability to generalize, confirming the results of Jelassi et al. (2023). The final total accuracy (correctness of the answer) reaches 0.93 after 1000 epochs.

Let us now repeat the analysis of the attention patterns, the PCA of the residual stream and the function of the (final) MLP. The attention patterns are shown in Fig. 23. We clearly see the staircase patterns we have been advocating for, and they are consistently of length six; much clearer as in the unprimed case in Fig. 21. We also see that in head 0:0, the previous sums are already calculated (in parallel) and stored in the integers in the input. In head 1:1 we also see that the previous sums get transferred to the current sum, similar to the two-layer case trained on just 3 digit addition.

Moving on to the PCA of the residual stream, given in Fig. 24, we again see the same structure as before. The plot shows the residual stream for the last six positions, which, according to the attention patterns in Fig. 23, is where the action happens. Let us focus first on the task division, i.e. the first four rows of Fig. 24. We studied five tasks labelled according to the same trinary system we used

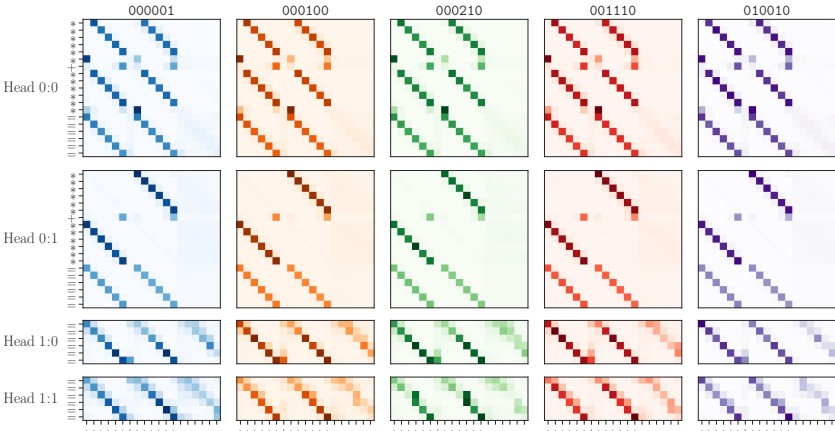

Figure 23: Attention pattern for 5 out of the 89 different cases for 6 digit addition in case of the primed model. The cases are labelled using a trinary system, where a 0 means $< 9$ at this position, 1 means $> 10$ and 2 means equal to 9. Interestingly, the attention patterns look similar to those in Fig. 21, but with some crucial differences. First, the attention patterns are much clearer (the model is more confident in which digits need attention). Second, head 1:1 still transfers some information from the previous sum to the current one, but it appears head 0:0 is doing most of the heavy lifting for that.

to label the attention patterns in Fig. 23 with. We notice that the first layer (first two rows) exhibit a separation between whether the summed digits result in $\geq 10$ or not, similar to our findings in Sec. 4. In the second layer (third and fourth row), we observe that the model uses the leading principal axis to distinguish between examples that require a carried one or not, again as we saw in Sec. 4. The last four layers are labelled according to the target digit at that position, but there is not a lot of structure other than that after the final MLP, where the examples are ordered according to numerical value of the target digit.

It is also interesting to note that the PCA and attention patterns in the primed case are refinements of the unprimed case, where examples were sometimes only weakly separated and the attention was not as clear. This suggests that even these 100 six digit examples we added, can help the model to generalize to larger length, essentially by generalizing the existing structure required for 3 digit addition that was present in the model from early on.

Finally, let us consider the carrying of the one. If we look at the final MLP and ablate it, then it does not seem to have a clear role for adding any carried one. This is perhaps expected, because the attention patterns suggest something more intricate. Nevertheless, we can still reverse engineer this part. As we see from the attention patterns, head 0:0 and head 1:1 seem to be important for carrying the one. From the discussion in Sec. 4 this suggests the final MLP is perhaps important for carrying of the one. As we already mentioned, this is not the case, at least in the way we saw in Sec. 4. Instead, when we ablate the final MLP, all sums are reduced significantly in accuracy, except the carry sums in the first three positions. So the final MLP is responsible for correcting the non-carry sums. For head 0:0, an ablation study suggests it is important for the carrying of the one in the first three positions (the OOD positions). So despite the carrying of the one is a bit scattered throughout the network in this case, we are still able to find parts of the network that are important for this operation.

To get more insight into what the final MLP does do, we look at an SVD of the pre-activation weights as discussed in Sec. 5. This shows us that there is again a division between neurons that are important for carrying the one, similar to what we saw in Fig. 13 and 14. So despite ablating it gave a different effect on the accuracy, the pre-activation weights' leading principal axis (or a linear combination of the two leading ones) still represents the *carrying over* feature.

The pieces of evidence we have presented are thus giving us a implementation of the carrying over algorithm that generalizes from three to six digits by using 100 priming examples. We also studied 3 to 10 digits with 500 priming examples, which gave analogous structures. In the decoder-only

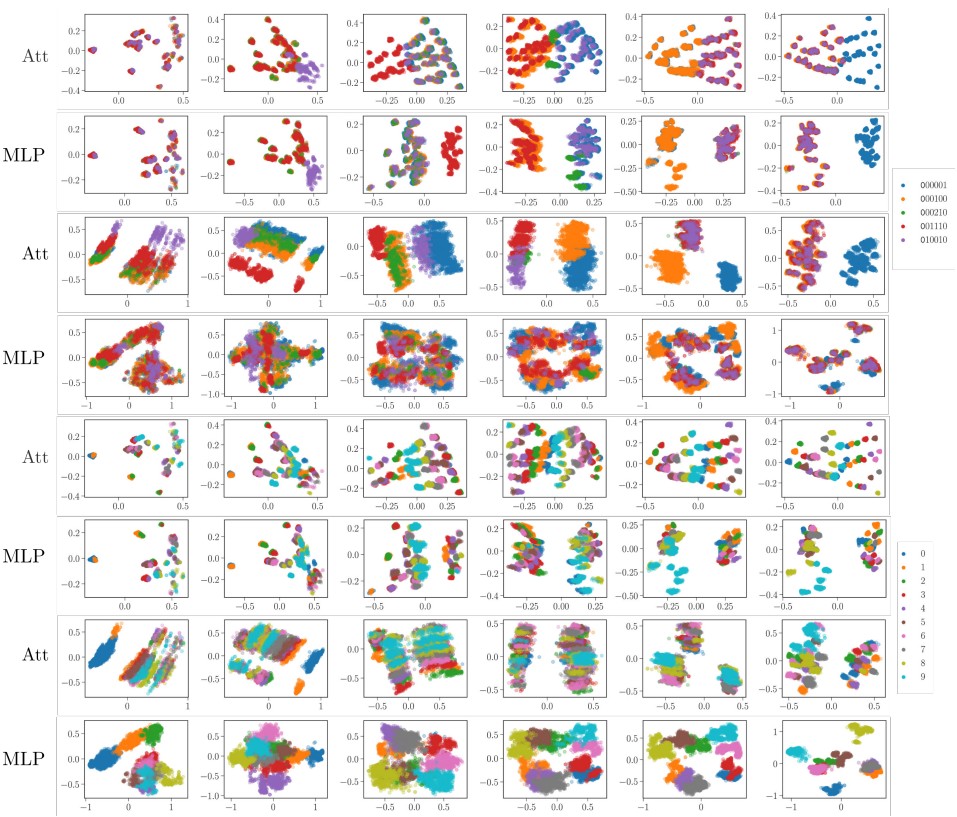

Figure 24: PCA analysis of the primed model for a set of 20k examples for 5 tasks, which are again labelled by our trinary system. Again, we plotted the two leading principal directions for the six output positions (last six positions). In the first four rows, we see a clear division between tasks developing when the data traverses through the model. In the last four rows, the model orders data according to the target digit as can be seen after the attention in the first layer and the output after the final MLP.

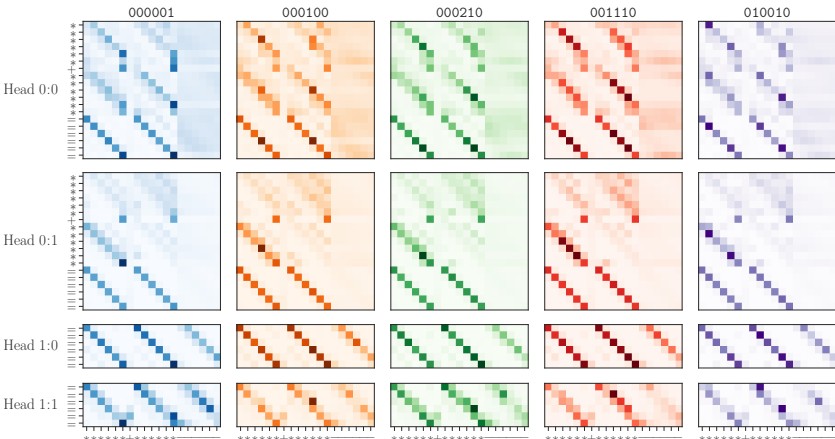

Figure 25: Attention pattern for 5 out of the 89 different cases for 6 digit addition in case of the model obtained by finetuning the unprimed model at epoch 500. The cases are labelled using a trinary system, where a 0 means $< 9$ at this position, 1 means $> 10$ and 2 means equal to 9. Interestingly, the attention patterns look similar to those in Fig. 21, more so than the primed model ones. We see head 1:1 still transfers some information from the previous sum to the current one, but so does head 0:0.

setting we also studied priming, but it is much less effective. Even for generalizing from 3 to 6 digits, 500 priming examples only increased the 6 digit accuracy to 0.59, but also hurt the 3 digit performance. Internally, the model does construct some generalizations of the length-3 structures, but we will leave a more detailed study to future work.

## H.2 FINETUNING

Besides training entirely new models with a primed dataset, one can also consider finetuning the unprimed model discussed before. This model has already the enlarged context window, so it only needs to learn 6 digit addition. We take 500 six digit sums and train the unprimed model (at epoch 500) with the same hyperparameters, but train for 50 epochs[8]. The finetuned model reaches 0.94 on six digit sums and 0.97 on 3 digit sums. The finetuned model also stayed close to the original model, since we found the weight norm only changing by 0.2%, from roughly 481 to 482.

If we consider the finetuned model's internals, we see a very similar implementation as before. Apart from the final MLP carrying the one, we see the same structure as we discussed in Sec. 4. For instance, the attention patterns are given in Fig. 25 and look very similar to the ones in Fig. 21, but the staircases and information transfer from the previous to current sum are much clearer. The structure of the hidden states is also again such that at layer 0 it is separated according to whether the sum of digits is $< 10$ or $\geq 10$ at a given position and at layer 1 according to whether a carried one is needed or not. This separation is much cleaner than in Fig. 22. Furthermore, despite ablating the final MLP not giving a clear reduction in accuracy of the carry sums, the SVD analysis shows a feature dimension corresponding to *carrying over*, just as we saw before.

## I ANALOGY WITH ELECTRICAL CIRCUIT

It is interesting to compare the two layer models with what in electrical engineering is called a *full adder*. This is a digital circuit that performs binary addition with carrying over, for instance see Sec. 4.3 in Morrison Mano (1979). One way of thinking about this is in terms of half-adders, which are circuits that add two binary entries using an XOR gate and compute a carried one using an AND gate. So they have two inputs and two outputs. Concatenating two of them results in the addition of

---

[8]We use the same batch size, so each step is just going over all 500 examples already. We also used batch size 10 and trained for 10 epoch, but that gave similar results. It would be interesting to study this dependence in more detail.

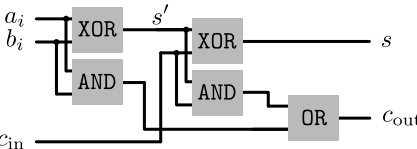

Figure 26: Digital circuit schematics of a full adder as two half-adders.

two binaries with a carried one from any previous positions. For instance, see Fig. 26. Here $a_i$ and $b_i$ are the inputs to be added at some position $i$, $c_{\text{in}}$ is the carry info from the previous position. $s$ is the output sum and $c_{\text{out}}$ is the output carry info. It is straightforward to construct a truth table for the full adder. We can make a rough comparison between this circuit and ours. The first half-adder (first XOR and AND gate) is equivalent to the first layer, the final MLP is the second XOR gate and the rest the second attention block as it is responsible for transferring the carry information to the next position. This comparison highlights a few things. 1) the need for two interactions between different positions. 2) ablating the MLP is equivalent to replacing the XOR with a direct connection between $s'$ and $s$. 3) ablating the *decision head* is as if the XOR gate is replaced by a *leaky* version that still gives 1 whenever any input is 1 but whenever both inputs are 0 it sometimes gives a 1 or a 0 as output. This preserves the ability to do the calculation whenever a one needs to be carried but gets confused otherwise.

