# OpenReview forum: "Carrying over Algorithm in Transformers"
_ICLR.cc/2024/Conference — Submitted to ICLR 2024_

### Official Review · Reviewer_RdPh · 2023-10-23

**Soundness:** 2 fair
**Presentation:** 2 fair
**Contribution:** 2 fair
**Rating:** 5
**Confidence:** 4

**Summary:**

In this paper, the authors study whether simple Transformer architectures might learn a carrying over algorithm to solve arithmetic tasks. To this aim, they train and test two-layer Transformers on a 3-digit addition task, and show that the emerging attention patterns allow the model to solve the task in a modular fashion, by dedicating some resources to the simpler sub-task of digit summation and other resources to the task of deciding whether carrying over is required, and in such case to actually perform carrying over. The authors argue that their findings could extend to Large Language Models, such as Alpaca 7B, and provide “suggestive evidence” in support of this claim.

**Strengths:**

- The article is overall well-written and well-organized.
- Understanding the inner working of Transformer architectures is certainly an important and timely research topic, especially for tasks related to “symbolic reasoning”, where often these models fall short. The ICLR community is particularly interested in such topic.
- Focusing on simple architectures and simple symbolic domains, such as arithmetic, allows to get useful insights that might then be extended to more complex architectures and tasks.
- The authors share source code (though I quickly checked it, and it does not seem to be properly commented / explained). This should guarantee reproducibility of results, but I encourage the authors to improve the code structure and provide more complete documentation.

**Weaknesses:**

- Many related works have been omitted.
- While I agree that digit arithmetics constitutes a valid testbed to build mechanistic accounts of the inner workings of deep learning models, the current task setting seems oversimplified. To establish the emergence of a putative carrying over algorithm, the authors should demonstrate that such algorithm allows to systematically solve the target problem, for example by testing the model over problem instances involving longer digits and more operands than those observed during training.
- The authors frame their research as an investigation of mathematical reasoning abilities of LLMs, arguing that a fruitful approach is to study simplified architectures and then extrapolate the results to larger models. While I agree that this might be a reasonable approach, the authors provide only “suggestive evidence” that their findings extend to one LLM (Alpaca). I think it would be more effective to directly frame this contribution as an investigation of the computational capabilities of simple Transformers architectures, and mention as a side point their putative application to LLMs.
- There are a few methodological details than require clarification (see questions below).
- English phrasing and grammar could be improved.

**Questions:**

- The “Related works” section only mentions a few studies related to the emergence of grokking in Transformer architectures. This section should be extended by including relevant papers that have investigated the capability of Transformer architectures to learn arithmetic problems. For example, others have proposed how to improve the standard Transformer architecture to improve generalization on simple arithmetic tasks [1], possibly by exploiting grid-like problem representations that promote the emergence of carrying over algorithms [2]. More recent work has also studied how systematic length generalization in basic integer arithmetic might be improved using relative position embeddings [3].
- To demonstrate that Transformers are learning an algorithm, their behavior should be tested also with out-of-distribution problem instances (as usually done in the related literature, see papers mentioned above). This would include longer operands than those used during training (> 3-digits) but also more operands than those used during training (> 2, in this case).
- A related point is the issue of memorization. The authors should better explain why they believe that one-layer models are memorizing the solutions, while two-layer models are actually learning a systematic behavior.
- The authors argue that “Usually the carrying over algorithm is employed in a right to left fashion”. This is the “human way” to implement the carry over algorithm, which might be due to processing constraints (we prefer to process things sequentially when the problem requires deliberate reasoning). This might not apply to Transformers, which should learn to parallelize the computation whenever possible (also see [2]) In line with this observation, it would be interesting to better investigate model behavior on the “Non-carry” (NC) scenario (subset #1). In such case, I would expect the Transformer to execute the entire operations in parallel.
- What is the rationale for using three = at the end of the arithmetic operations, rather than just one?
- “The attention patterns are averaged over the test dataset”: shouldn’t the authors average only similar cases, to avoid mixing carry vs. non-carry cases (or even according to subsets identified by the authors at pg. 3)?
- Phase transitions: from Fig. 2 it is not evident that phase transitions are actually occurring. The figure only shows that at some point the attention pattern changes, but in order to be a phase transition such change should happen suddenly (e.g., between two consecutive epochs). Furthermore, following the authors’ narrative this should happen not just during two consecutive epochs, but during the epochs corresponding to the abrupt changes in the loss function. From the figure, this does not seem to be the case. Can the authors clarify this?
- In the figures, the curves representing trends of just one individual run should be replaced with curves representing mean and variance (or min-max) of the six multiple runs.
- Extending the present findings to LLMs is not easy, though the anecdotical evidence provided by the authors suggests that there might be some commonalities. Since a major limitation of Alpaca under the setup investigated by the authors was the overall low accuracy, it would be interesting to study whether prompting techniques could improve model performance, and maybe promote the emergence of a more interpretable algorithmic strategy. Also testing other LLMs might allow to establish whether more accurate models would use more systematic strategies at inference time.


References
1.  Csordás R, Irie K, Schmidhuber J. The Neural Data Router: Adaptive Control Flow in Transformers Improves Systematic Generalization. ICLR, 2022.
2.  Cognolato S, Testolin A. Transformers discover an elementary calculation system exploiting local attention and grid-like problem representation. IJCNN, 2022
3. Jelassi S, D’Ascoli S, Domingo-Enrich C, Wu Y, Li Y, Charton F. Length Generalization in Arithmetic Transformers. http://arxiv.org/abs/2306.15400, 2023

---

> ### Author Response · Authors · 2023-11-15
>
> We thank the referee for the review and comments!
>
> We will update the source code for better reproducibility.
>
> Reply to weaknesses:
>
> 1) See our reply to your first question.
>
> 2) It is true that exploring the task for larger integer addition is a very important one and that the carrying over algorithm works for integer addition of any length. As we mentioned in appendix G, the model trained on 3 digit addition does badly on 5 digit addition. However, we believe this does not mean the carrying over algorithm is not implemented, it just means that the carrying over algorithm is implemented for 3 digit addition and partly for 5 digits. To be more specific, the model can still do the summations at each digit and carry a one, but gets 'confused' when more digits are present. We will add a discussion about longer lengths to the paper.
>
> 3) We agree that the mentioning of LLMs is mostly motivational and study of Alpaca 7B provides suggestive evidence. Clearly more study is necessary to make a more concrete link between the small and large models. See also our response to your last question.
>
> 4) Regarding the English phrasing/grammer, we will go over the text again to fix that.
>
> Reply to questions:
>
> 1) We thank the referee for mentioning these other references, they look indeed very interesting for studying ways to improve the standard Transformer architecture. However, we would like to mention that in our work, we were not necessarily focussing on improving the standard Transformer (we only included RoPE) as to the best of our knowledge many much larger models still use the standard Transformer (and often with a rotary embedding). This might be the reason why Refs. [1, 2] have not caught our attention, but certainly we should have included Ref. [3].
>
> 2) We agree with this comment and will add some additional discussion to the paper.
>
> 3) We thank the referee for brining this up, the memorization discussion in the appendix B just addresses the one-layer model, which does not learn the addition problem perfectly. It does however, do the summation of the digits. What we meant with memorization here is just the fact that the residual stream seems to have large spread and not many localized pockets as we saw in the two-layer model. This indicates that there is not a special subspace in the residual stream dedicated to a particular feature of the data (like carried one necessary or not), but rather it is just treating each example differently. However, we do notice some interesting ordering, so it is also not just memorizing either. Perhaps memorization is not the correct terminology here, we will rewrite this part accordingly. The 2-layer model learned something systematic, as we illustrated in Fig. 1, because i) the residual stream is much more divided up into pockets related to specific feature of the data. ii) 2-layer models 'understand' where to add a carried one (from the attention patterns and the residual stream in the second layer) and iii) also have a particular component (the final MLP) that does that actual carrying of the one. We will try to clarify this further in the paper.
>
> 4) The referee is absolutely correct here, we indeed meant here the human way and did not try to make any claim the Transformer should or should not do this. The parallelization is something we also see from the attention patterns. They provide the addition of the digits all in parallel. We thank the referee for mentioning this parallelism, which we have incorporated in sections 3 and 4. Any carried one is added later in final MLP of the two-layer network. This is indeed particularly clear in the non-carry task.
>
> 5) For the encoder-only models, the rational for the three equal signs was to read off the answer above these positions. One could also have done different things, for instance not use any equal signs at all, but we have not experimented with other choices.
>
> 6) Here we were interested in the overall structure of the attention patterns, i.e. whether the staircase pattern would develop or not and present the reader with the transitions. This did not require worrying about the different tasks, which was relevant for two or more layers, i.e. figure 3.
>
> 7) Here we plotted the attention patterns within each phase and checked that indeed the pattern suddenly changes whenever the loss drops (or accuracy increases suddenly). We will clarify this a bit more in the main text.
>
> 8) We thank the referee for this comment, but the bold curves are the mean and so the variance can be roughly deduced from the plots since there are not that many. We can adjust this to the referee's suggestion.
>
> 9) We thank the referee for mentioning this. We agree this is indeed a limitation. As we have discussed in replies to other reviews, we have tested our hypothesis on llemma-7b (https://arxiv.org/abs/2310.10631) as well. We will discuss this further in the paper and hope this addresses the question of the referee.

---

> > ### Comment · Reviewer_RdPh · 2023-11-17
> > **The revised paper might improve on several weak points, but not on the most critical one.**
> >
> > I thank the authors for having considered my comments and for their thoughtful reply.
> >
> > I think that several of my concerns could be fixed by implementing the actions mentioned by the authors (e.g., uploading the source code, fixing language and grammar, mentioning some extra references, adding results from other LLMs, improving the discussion...).
> >
> > However, I still believe that the paper has some critical weaknesses that would not be addressed according to the planned revision. When we talk about "learning an algorithm" (such as the carrying over algorithm), we mean that the model learns a systematic procedure that can be applied to inputs of arbitrary complexity (indeed, the focus of algorithmic learning papers is usually on evaluating generalization capabilities). The current evidence suggests that Transformers can learn a procedure to solve 3-digit addition, but that does not allow to systematically generalize to additions with more operands and I think this is a key limitation of the current study. The authors argue that ”this does not mean the carrying over algorithm is not implemented, it just means that the carrying over algorithm is implemented for 3 digit addition and partly for 5 digits". However, I wouldn't call this a "carrying over algorithm", since it does not generalize even to just a few extra operands.
> >
> > In conclusion, I still believe that although analyzing the internal mechanisms of Transformers in solving 3-digit addition is interesting, it does not constitute a sufficiently novel contribution to justify publication in a top-tier venue like ICLR.

---

> > > ### Author Response · Authors · 2023-11-20
> > >
> > > Thank you for your comments and thoughtful reply!
> > >
> > > We have updated the revision and hopefully it addresses your comments.

---

> > > > ### Comment · Reviewer_RdPh · 2023-11-21
> > > > **Question about Fig. 20**
> > > >
> > > > Can you please clarify what is represented in Fig. 20? The figure has six subplots, but their meaning is not explained neither in the figure caption nor in the text (unless I missed something).

---

> ### Author Response · Authors · 2023-11-21
>
> Thanks for the question and sorry this was not clear. We have uploaded a revision that should clarify the confusion. In short, this is the test accuracy per position for the six digit sums.

---

> > ### Comment · Reviewer_RdPh · 2023-11-22
> > **The new manuscript improved compared to the initial submission**
> >
> > I thank the authors for having seriously considered the remaining issues that were not solved during the first round of revision. I appreciate their effort in trying to better highlight the novelty and the relevance of their contribution, and indeed I find the new analyses on 6-digit operations (with priming) interesting. I think it would be fairer to avoid claiming that the models considered "learn an algorithm" (maybe the authors can use a surrogate but less strict term, such as "procedure"?), since the current results do not allow to argue that the model discover a systematic way to solve arithmetic operations.
> >
> > I am still not convinced that this manuscript is original enough to be presented at ICLR, however I decided to raise my overall score from 3/10 to 5/10, and judge the contribution level to "2 fair" to reflect these improvements.

---

> > > ### Author Response · Authors · 2023-11-22
> > >
> > > Thank you for your comments, thoughtful considerations and increasing the rating! We are glad to hear that the additions are appreciated and you find them interesting. Your suggestion about changing the wording from 'algorithm' to 'procedure' is a good idea.
> > >
> > > We regret to hear you find the work still not original enough to be presented at ICLR. Let us try to convince you once more of why we believe it would be an interesting addition to ICLR,
> > > 1) The carrying over algorithm is a very simple algorithm that forms much of the basis of arithmetics. We believe that if we want to understand how transformers do arithmetics and how it can be improved, we should at least be able to fully understand integer addition. In the past this has been done, but to the best of our knowledge, a thorough discussion of small models (with the digit representation, which is relevant for LLMs etc.), their internals, including when priming or using finetuning is used, and their interesting and intuitive learned representations has not been done.
> > > 2) The added length generalisation discussion highlights that even though the model fails to generalize in terms of accuracy, the model's internal structure suggested many ingredients to actually do the generalisation are already in place. The finetuning discussion provides some evidence for this, but it definitely requires a more thorough investigation, in particular because it could help with improvements of the architecture. Nevertheless, we believe that that discussion is novel.
> > > 3) The updated discussion of applying our lessons from small models to a few LLMs (all trained differently) shows similar learned representations can show up, despite the model being many many times larger. This is particularly visible in the new Fig. 6, which shows striking similarities with the small models. A priori, we believe, that this 'generalisation' was not expected, because the model could do something rather uninterpretable. We think that the fact that this is potentially possible makes this an interesting and novel addition for the interpretability community and ICLR community as a whole.

---

> > > > ### Author Response · Authors · 2023-11-22
> > > > **Added disclaimer about 'algorithm'**
> > > >
> > > > We have updated the manuscript and added your suggestion as a footnote at the beginning of section 4 on page 4. We hope that this modification addresses your concern.

---

### Official Review · Reviewer_5YTw · 2023-10-28

**Soundness:** 3 good
**Presentation:** 3 good
**Contribution:** 3 good
**Rating:** 6
**Confidence:** 2

**Summary:**

In this work, the authors study how transformer models implement carrying over algorithm. They first focus on one layer and two-layer encoder-only models and show that the carrying over algorithm is implemented in a modular fashion, then apply the learned lessons to a large language model.

**Strengths:**

1. The proposed method is technically sound.
1. The empirical results look promising.
2. The paper is overall well presented.

**Weaknesses:**

1. Except for Alpaca 7B, the authors should provide more discussion for other transformer models in LLM and other modalities tasks.
2. The innovation, is limited as the main ideas have been circulated outside carrying over algorithm in Transformers. overall, beyond limited novelty, the paper has no major weaknesses. The limitation comes from the approached theme, while being promising, has yet to make an impact outside the research community. This means also, that the auditorium is limited.

**Questions:**

1. It would be better for authors to provide a more detailed discussion on large-digit addition (e.g. 5-dight, 6-digit, etc), since it can be a more complex task and may have different situations with 3-digit addition.
2. In this work, the authors analyze the attention pattern for 2 heads in each layer for one-layer and two-layer encoder/decoder-only models. It would be interesting to consider more heads for each layer in these small models.

---

> ### Author Response · Authors · 2023-11-15
>
> Thank you for the review and comments!
>
> Reply to weaknesses:
> 1) We thank the referee for bringing this up. We agree that other modalities would be interesting to study. However, this would be particularly interesting for studying the training dynamics or if the different tasks have something in common, how they affect the implementation of any algorithm. The focus of this work was more on the final trained model instead for a single task. We believe the task considered here is already somewhat rich enough. However, we do believe that testing on other LLMs is important, so we have added also tested our hypothesis on llemma-7B (https://arxiv.org/abs/2310.10631). See reply to the other reviews. We will incorporate this in a revision.
>
> 2) We thank the referee for the vote of confidence here! We believe the novelty is there (see also replies to other reviews), but indeed it is a specialized setup, but we believe still relevant for the ICLR community.
>
> Reply to questions:
> 1) In appendix G, we studied 5 digit addition when the model is trained on just 3 digit addition. This discussion is rather short and we agree with the reviewer it would be good to extend the discussion there and in the main text, which we will do in a revision. Notice also that the 4 digit case has already a larger number of different situations and is discussed in appendix F.
>
> 2) Thank you for the suggestion, that is indeed be interesting, but already in the two headed case, we saw there was only one head that was doing the heavy lifting and so we think increasing the number of heads would just isolate again a single head (in each layer) that would have the same job as in the two headed case.

---

> ### Comment · Reviewer_5YTw · 2023-11-22
>
> Thank you for your responses. That resolves most of my questions. However,  I would like to highlight two points. 1) There is a substantial distinction between various modalities. I still appreciate the comprehensive evaluation of the proposed method under different modalities such as images and texts and explore the commons across multiple modalities. 2) The current results suggest that the proposed method can learn a procedure to solve 3-digit addition, but it cannot generalize to additions with more operands well.

---

> ### Author Response · Authors · 2023-11-22
>
> Thank you for your comments.
> 1) We agree with you that indeed other modalities are very interesting, but the main focus of this paper was integer addition and understand that task in various ways. We are not sure what you mean with other modalities in this context? What are you precisely suggesting? Perhaps you mean reverse-engineering in general and see whether you can get insights from looking at the internals of transfomers? Regarding that, many works discuss this, but we cited Nanda et al. here because they focus on the combination between arithmetics and interpretability. We also cited the works Voss et al., and Elhage et al. as they discuss also in depth aspects of interpretability of transformers and how to understand the model's activations (which we used in the SVD analysis in section 5.).
>
> 2) We believe that in the revisions we have made, we have addressed a bit the generalisation to larger length. As you mention, just training on 3 digits will not generalize, which we believe was known. However, as discussed in Jelassi et al., generalisation can be improved by priming or finetuning. We explored this and found evidence that despite the accuracy not showing generalisation capabilities, the model's internals suggest otherwise and do seem to have all the ingredients for generalisation. Indeed, finetuning on just 500 six digit sums can give you near perfect generalisation in just 50 epochs and we believe this is because the model's internals are already such that they can do good at larger digits, they just needed a little push :). The same happens for priming, where we added 100 six digit sums to the original training set and show the trained models give near perfect generalisation. We hope that with this additional discussion in the revision, we addressed your point in a satisfactory manner.
>
>
>  For instance, with priming or finetuning one can easily (and quickly) get them to go from doing 3 digits well to say 6 digit sums. See section 6 and appendix H for the details.

---

### Official Review · Reviewer_tJZK · 2023-11-01

**Soundness:** 3 good
**Presentation:** 3 good
**Contribution:** 2 fair
**Rating:** 5
**Confidence:** 4

**Summary:**

The objective of this work is to understand the nature of arithmetic operations as performed by transformers.  Specifically, the paper focusses on 3-digit addition with carryover as the use case.   The main results are in the nature of attention patterns and MLP weights that evolve as the transformer is trained.

**Strengths:**

The paper raises an interesting question of how the attention of a transformer and the MLP weights evolve to correspond to its performing arithmetic.   The paper is clearly written in general.

**Weaknesses:**

At some level, the paper's contribution is not in the transformer training itself, which follows on standard lines, but is on interpreting the outcome in terms of structure of attention and the MLP layers.   My general opinion is that, while interesting, this is not a sufficient contribution in itself.  For instance, the paper does not offer any major prescriptions on how best to train a transformer for enabling it to do arithmetic more accurately, nor does it provide an analysis of why the condition distribution generated by a transformer has the correct structure to perform carry over arithmetic.

**Questions:**

I do not have any technical questions.  However, I am concerned at the level of novel contribution of this paper.

---

> ### Author Response · Authors · 2023-11-15
>
> Thank you for the review and comments!
>
> You are right that the emphasis of this work was on the analysis of the final trained model, not necessarily on the training itself. Some of the structures we found in the final model prompted us to study the training in more detail, i.e. the transition discussed in section 3 and the evolution of the MLP.
>
> Let us address the reviewer's worry about novelty:
>
> 1) We have indeed not addressed ways to improve the accuracy. We agree this is indeed a very interesting and crucial line of research, but was not the focus of this work. We focussed on studying an ordinary and commonly used transformer architecture and study it in depth. We give a detailed account of the attention patterns and why they make intuitive sense, we study the learned representations in the residual stream and show it has some interesting and interpretable structure, we study the final MLP in detail and discover a clear role for the carrying of the one. We were able to do so for a variety of hyperparameters. As far as we are aware, we have not seen such an analysis that includes not only just the attention heads but also the MLP and their development during the course of training.
>
> 2) One could say that the knowledge obtained by studying the network in the way we did, could be used to understand how to improve the accuracy on OOD data (i.e. larger length). While we agree this is indeed interesting and important, we have decided to go a different route and understand generalization of the lessons learned in small models to much much larger models. Often this could be a rather futile exercise, but we believe the novelty here is that we provide concrete signs that a 7B LLM has a similar implementation as we saw in the small models. In particular, i) we see the same attention patterns, ii) Only two types of heads are needed: one responsible for adding the digits, and another for transferring the summed digits to the output (we want to reiterate that this is not necessarily expected, because the model could do this in many different ways), iii) Part of the residual stream has a similar structure.
> Clearly these are signs and not hard evidence, and more research is definitely necessary to be more conclusive. To alleviate this slightly, we have also tested another LLM, llemma-7B (https://arxiv.org/abs/2310.10631). This model is supposed to be good at math, and indeed for the arithmetic task we find 86.7% accuracy on 1000 4 digit sums. Just as in Alpaca, we can identify two heads that are most important as ablating those causes the accuracy to drop to 30.0%. We will put this more clearly in a revision.
>
> At any rate, we believe that the combination of 1) and 2) is novel and having these types of examples are potentially motivating as (mechanistic) interpretability of small models can suffer from this kind of generalization and it is not often one is in the position of seeing such signs of lessons from small models in much much larger ones.

---

> > ### Comment · Reviewer_tJZK · 2023-11-21
> >
> > Thank you for your response.  I would like to compare your approach to Gregory Valiant (see his Simons Foundation talk on in-context learning on YouTube), in which he makes a strong case that transformers can learn linear regression and perform in-context estimation tasks in general.  The point there is to understand the nature of the conditional distribution generated by the transformer.   The approach in this paper is almost like probing the human brain to see which portions of it fire for speech, which ones for movement etc.  But the transformer is an engineered system -- I  am not convinced that identifying the heads for different subtasks gives us any prescriptive solutions for better engineering or using the architecture.

---

> ### Author Response · Authors · 2023-11-21
>
> Thank you for your comment. With the revisions we made, however, we hope that it does address some of your earlier points.
>
> Regarding your comment: We believe that studying the internals of transformers is a useful endeavour precisely for the reason you are alluding to. If one not only understands the conditional distribution generated by the transformer, but also how it is generated, i.e. what components of the attention layers generate what features in the distribution, one can try to improve the architecture or adapt them so that the generated distribution is closer to what one wants. As we mentioned before, we have not focussed on the 'improving' aspect, but that is certainly an interesting question.

---

### Official Review · Reviewer_rE6h · 2023-11-04

**Soundness:** 4 excellent
**Presentation:** 4 excellent
**Contribution:** 4 excellent
**Rating:** 10
**Confidence:** 4

**Summary:**

1. Authors proposed a transformer models that implements carrying over algorithm on one,two and three layer model.

**Strengths:**

1.The novelity of the paper is excellent.
2.Sufficient results are presented and their analysis is carried out.

**Weaknesses:**

No

**Questions:**

1.Results of the figure 13 need to be cross verified.

---

> ### Author Response · Authors · 2023-11-13
>
> Thank you for your review and comments! What do you mean with the results of figure 13 need to be cross verified?

---

### Author Response · Authors · 2023-11-20
**Revision**

We want to thank the referees again for their thorough reading of the manuscript and their thoughtful comments.

It seemed that in general the referees had two bigger comments: length generalisation and other LLMs. We will address these comments first and then move on to other revisions we made.

For length generalisation and other LLMs, we have made a big revision to section 6. Essentially it now contains two parts, one on length generalisation and another where we focus on integer addition in three LLMs, Alpaca 7B, Llemma 7B and Zephyr 7B. As a result we also changed a bit the phrasing in the introduction and tried to be more cautious in our phrasing. To be a bit more specific:

1) We agree with the referees that the carrying over algorithm is a 'length independent' and so it should work for different lengths as well. We therefore studied generalisation from 3 to 6 digits (and 3 to 10 as well). We added padding to the inputs and trained a few models and found them to not generalize to larger length, which was probably expected. However, we found that despite the low accuracy on six digit sums, all ingredients for a carrying over algorithm are in place. This suggests that priming is an ideal playground to force generalisation (the reference brought up by the referee was indeed known to us, but it is unclear why we did not mention it in our earlier version). We primed with 100 6 digit sums and found the models to generalize almost perfectly and with a rather clear implementation of the carrying over algorithm. We mention these results in section 6 and study and compare the primed and unprimed models in a new appendix, App. H. (the previous appendix G on 3 to 5 digit generalisation is removed)  We also provided the code at the same github repo for reproducibility.

2) The referee's were right that the performance of Alpaca is rather low and thus hard to draw conclusions from. As advertised, we added a discussion of other LLMs to section 6. The two LLMs we added are Llemma 7B and Zephyr 7B, both of which are rather good at our addition task and so one might have more success in seeing (part of) a carrying over algorithm. Indeed, by studying the attention patterns and residual stream (see github repo), we find important heads which have the staircase patterns and see the same separation in hidden states in the residual stream as we saw for the smaller models. In particular, see the new Fig. 6. We moved the discussion of Alpaca 7B to appendix G.

We hope that these revisions address the referee's comments about length generalisation and the study of other LLMs.

Other revisions we made are:
1) We improved the related works discussion and included more relevant references. We regret we did not include the suggested literature.
2) We improved the overall discussion by rewording things and fixing grammar/typos.
3) We improved the discussion on memorization in one-layer models by comparing directly to the two-layer models, as we already mentioned in the reply below. We did this in both the main text and appendix B.
4) Cleaned up the githup repro, but will continue to do so till the deadline to ensure reproducibility.

We hope these other revisions address the referee's comments.

---

### Author Response · Authors · 2023-11-21
**Revision v2**

We want to thank the reviewers again for their thoughtful and insightful comments and questions. We have submitted a new revision in which we changed:

1) We fixed some typos and things that were not clear in the previous revision.
2) We expanded a bit on the priming in appendix H.1, we included a discussion on the SVD analysis of the final MLP, showing it again has a feature dimension associated with carrying over.
3) We added a paragraph about finetuning as discussed in the Jelassi et al. paper. We find that finetuning of models that are trained on 3 digit but with padding to allow for 6 digit sums also helps with generalisation and the implementation of the putative carrying over algorithm is similar. See appendix H.2 and section 6 in the main text.

We have also updated the repository https://github.com/CarryingTransformers/CarryingTransformers so as to ensure the added things in the revision are reproducible. We also added comments etc. to the codes to train our models.

---

### Meta-Review · Area_Chair_tEJq · 2023-12-05

**Metareview:**

This work studies the mechanisms of a trained two-layer Transformer on 3-digit addition problems. By experimentally examining the attention and activation patterns, they argue that Transformers implement a specific algorithm for addition (involving parallel sums and carrying).

Reviews were borderline, with two weak-rejects and one weak-accept. (Unfortunately Reviewer rE6h’s response does not constitute a review, and must be discounted).

Reviewers noted that this work studies an important and timely question, towards understanding the abilities of language models on symbolic/reasoning tasks.
However, reviewers were not enthusiastic about the strength and novelty of the results. The proposed setting is already very simplified (3 digit addition, on 2 layer models), but the results are not precise or insightful enough even in this setting.
Specifically, as noted by reviewers, the experiments are somewhat consistent with the algorithmic hypothesis, but they are far from conclusive. A few examples:
* The trained Transformer does not generalize to longer inputs, for reasons that are unclear. The authors have added further discussion in the Appendix, which is appreciated, but it is unclear how to interpret these results as consistent with the main message.
* A number of claims are made informally, and assumed to be true, without justification. For example, in defining the candidate algorithm, the authors write “the beauty of this dataset is its natural division in subsets...”. However, as Reviewer RdPh also notes, this is a division which is “natural” for humans, but not necessarily for Transformers (indeed, it is unclear what “natural” means for Transformers).
* The authors make various conclusions by interpreting PCA of activations. It is not clear why this methodology is justified.

In general, the paper currently provides heuristic evidence that is only weakly supporting the main hypothesis (which is a strong hypothesis). Thus I must recommend rejection. I advise the authors to consider the feedback from all reviewers.

**Justification For Why Not Higher Score:**

The evidence is only weakly related to the hypothesis; reviewers do not consider the contribution to be strong enough for ICLR.

**Justification For Why Not Lower Score:**

N/A

---

### Decision · Program_Chairs · 2024-01-16

Reject